# Insights in Molecular Therapies for Hepatocellular Carcinoma

**DOI:** 10.3390/cancers16101831

**Published:** 2024-05-10

**Authors:** Philipp Heumann, Andreas Albert, Karsten Gülow, Deniz Tümen, Martina Müller, Arne Kandulski

**Affiliations:** Department of Internal Medicine I, Gastroenterology, Hepatology, Endocrinology, Rheumatology, and Infectious Diseases, University Hospital Regensburg, Franz-Josef-Strauß-Allee 11, 93053 Regensburg, Germanykarsten.guelow@klinik.uni-regensburg.de (K.G.); deniz.tuemen@klinik.uni-regensburg.de (D.T.);

**Keywords:** hepatocellular carcinoma, targeted therapy, molecular directed therapy

## Abstract

**Simple Summary:**

Molecular directed therapy for hepatocellular carcinoma (HCC) involves targeting specific signaling pathways that normally promote cancer cell growth and survival. Accordingly, a blockage of these pathways leads to tumor shrinkage and improves the patient outcome. Recent advancements in molecular directed therapies have focused on tyrosine kinase and angiogenesis inhibition, thus targeting pathways involved in the molecular pathogenesis of HCC. Especially the vascular endothelial growth factor (VEGF) signaling pathway is one of the most prominent pathways involved in HCC progression. The combination of inhibiting VEGF signaling and immune-directed therapy or double immunocheckpoint inhibition are the mainstay of contemporary treatment strategies and promising agents for future combination treatment regimens. However, the effectiveness of these therapies may vary among patients, highlighting the need to identify the specific molecular alterations for tailored therapeutic approaches. Additionally, in modern systemic therapy of patients with HCC, the underlying pathology of the liver has always been encountered to identify population subgroups in terms of pathophysiology and underlying liver function to predict response to different therapeutic regimens.

**Abstract:**

We conducted a comprehensive review of the current literature of published data and clinical trials (MEDLINE), as well as published congress contributions and active recruiting clinical trials on targeted therapies in hepatocellular carcinoma. Combinations of different agents and medical therapy along with radiological interventions were analyzed for the setting of advanced HCC. Those settings were also analyzed in combination with adjuvant situations after resection or radiological treatments. We summarized the current knowledge for each therapeutic setting and combination that currently is or has been under clinical evaluation. We further discuss the results in the background of current treatment guidelines. In addition, we review the pathophysiological mechanisms and pathways for each of these investigated targets and drugs to further elucidate the molecular background and underlying mechanisms of action. Established and recommended targeted treatment options that already exist for patients are considered for systemic treatment: atezolizumab/bevacizumab, durvalumab/tremelimumab, sorafenib, lenvatinib, cabozantinib, regorafenib, and ramucirumab. Combination treatment for systemic treatment and local ablative treatment or transarterial chemoembolization and adjuvant and neoadjuvant treatment strategies are under clinical investigation.

## 1. Introduction

The incidence of malignant liver tumors has been gaining increasing significance over the recent decades. Globally, primary liver cancer ranks as the sixth most frequently occurring cancer [1]. In 2020, 905.677 new cases of malignant primary liver tumors were reported worldwide, accounting for 4.7% of all diagnosed carcinoma in adults [2]. Among primary liver tumors, hepatocellular carcinoma (HCC) is the most common tumor entity, comprising 75–85% of cases [2]. HCC carcinogenesis arises from hepatocytes [3]. Rare malignant primary liver tumors include soft tissue tumors originating from connective tissue or blood vessels (such as sarcoma, hemangiosarcoma), embryonal tumors like hepatoblastoma, and fibrolamellar carcinoma. The mortality rate of primary liver tumors ranks second (830.180 deaths per year worldwide) after lung tumors, or third when considering colon and rectal carcinoma together [2]. The poor prognosis is evident from nearly equal incidence and mortality rates. With a 5-year survival between 14% and 21%, liver cancer is characterized by a poor long-term outcome [4,5]. Besides liver transplantation, radical surgical resection remains the only curative therapeutic option, although recurrences are very common. Consequently, adjuvant therapeutic concepts have been established in cholangiocarcinoma (CCA) and are recommended by international guidelines [6]. Adjuvant treatment after resection of HCC is also being investigated but is not the current standard of care. Early symptoms of HCC are often absent, leading to late detection, with most patients diagnosed at an advanced stage when surgical resection is unfeasible. In these advanced stages, oncological systemic therapy represents the current standard of care.

Molecular directed therapy has emerged as a promising approach in the treatment of HCC, contributing to significant advancements in patient care and addressing specific molecular pathways involved in HCC progression and proliferation. Drugs such as sorafenib, lenvatinib, regorafenib, and cabozantinib inhibit angiogenesis and tumor growth by targeting vascular endothelial growth factor receptor (VEGFR) and other signaling pathways. Immune checkpoint inhibitors (ICIs) like atezolizumab (in combination with bevacizumab), durvalumab/tremelimumab, ipilimumab/nivolumab, and pembrolizumab have demonstrated efficacy in HCC treatment by enhancing the immune response against cancer cells. They target programmed cell death protein 1 (PD-1), programmed death-ligand 1 (PD-L1), and cytotoxic T-lymphocyte-associated protein 4 (CTLA-4) pathways. Combining molecular directed therapies with other modalities such as chemotherapy, locoregional therapies (radiofrequency ablation, transarterial chemoembolization, radiation therapy) are being explored to enhance treatment efficacy and overcome resistance mechanisms [7].

Given the escalating worldwide incidence of HCC, further investigation is imperative to develop new (molecular) therapeutic strategies and enhance the patients’ prognosis. Continued research and clinical trials are essential to optimize the efficacy and safety of these therapies and to further expand their role in HCC. HCC is a heterogeneous disease with diverse molecular profiles and not all patients respond equally to treatment. Additional studies can elucidate which patient subgroups are most likely to benefit from specific molecular targeted therapies based on their molecular characteristics. A personalized approach can improve treatment outcomes by matching patients with the most effective therapies for their individual tumor biology.

This review aims to provide an overview of established molecular pathological therapeutic approaches in HCC and offer insights into new agents for the future. The study also addresses the current revision of the BCLC classification and addresses new approaches in both adjuvant and palliative therapeutic settings.

## 2. Risk-Factors and Diagnostic Algorithm of HCC

In general, most diseases lead to the development of liver cirrhosis and pose an increased risk of developing HCC. Chronic hepatitis B infection by genome-integration and non-alcoholic fatty liver disease are particularly associated with an elevated risk of HCC, even in non-cirrhotic condition [8,9]. Baseline factors, such as age, sex and comorbidities significantly influence the clinical outcomes of patients with HCC. Also, the presence and severity of cirrhosis significantly affect HCC prognosis. Cirrhosis is associated with liver decompensation, portal hypertension, and impaired liver function, all of which contribute to poorer outcomes in HCC patients. Nevertheless, the overall health status of patients with HCC, including performance status (e.g., Eastern Cooperative Oncology Group performance status), influences treatment eligibility and prognosis [10,11]. Regarding the influence of etiology on treatment outcome, current real-world studies have not demonstrated a significant difference in overall survival for the use of atezolizumab/bevacizumab in patients with viral-associated HCC compared with alcohol-associated HCC and non-alcoholic fatty liver disease-associated HCC [12]. Nevertheless, in a recent meta-analysis, it was shown that patients with viral-associated HCC are more likely to benefit from therapy with atezolizumab/bevacizumab, while patients with impaired liver function (Child Pugh B) are more likely to benefit from therapy with lenvatinib. In order to stratify therapy for specific subgroups, we believe further studies are necessary. Therefore, at present, therapy with atezolizumab/bevacizumab or durvalumab/tremelimumab is recommended as first-line therapy regardless of the underlying etiology (provided there are no contraindications to checkpoint inhibition) [13]. The common risk-factors for HCC and the estimated risk are reported in Table 1.

The diagnosis of HCC relies on contrast-enhanced imaging studies and/or histopathological analysis. Regular abdominal ultrasound is recommended for initial screening in all patients with liver cirrhosis [50]. Suspected HCC lesions should undergo further clarification through magnetic resonance imaging (MRI) and/or contrast-enhanced ultrasound (CEUS) [51,52,53]. In cirrhotic conditions, MRI has demonstrated superiority over contrast-enhanced computed tomography (CT) in HCC detection [54]. HCC lesions larger than 1 cm should be characterized based on typical contrast dynamics showing arterial hypervascularization and subsequent washout in the portal venous and venous phases of MRI (AASLD and EASL criteria for HCC diagnosis) [51,55,56,57]. If initial CEUS findings are inconclusive, complementary MRI imaging is recommended [50]. Clear contrast on MRI or CEUS allows for an HCC diagnosis without the need for biopsy in cases where curative treatment is intended.

However, in palliative treatment scenarios, even with definite cross-sectional diagnosis with typical contrast-enhancement, biopsy remains recommended to differentiate from mixed tumors and CCA and to further characterize molecular changes [14,50,57]. Complete tumor staging requires MRI of the liver and CT of the thorax. The Liver Imaging Reporting and Data System (LI-RADS classification), introduced in 2011, serves as the standard for interpreting findings from CT, CEUS, and MRI [58]. Yet, its validation on an appropriate patient population is pending. Positron emission tomography–computed tomography (PET-CT) is not currently recommended to confirm an HCC diagnosis [59]. Liver lesions smaller than 1 cm are challenging to distinguish from cirrhotic hyperplastic nodules [60,61]. Thus, close monitoring via ultrasound (recommended follow-up every 3 to 6 months by AASLD/EASL guideline) and complementary imaging for enlarging lesions is advised. Additionally, serum alpha-fetoprotein (AFP) determination aids in early HCC detection [62]. In AFP-high tumors, AFP serves as a progression marker for therapy response and early recurrence detection during follow-ups. AFP levels are also relevant for evaluating liver transplantation feasibility and decisions on systemic therapies [63,64], along with AFP dynamics for decision-making [65]. Staging of HCC also involves the assessment of liver function using the Child–Pugh criteria (total bilirubin, serum albumin, INR or prothrombin time, ascites, hepatic encephalopathy). Liver function critically influences therapeutic decisions, whether for surgical, radiologic-interventional, or systemic oncologic therapies. Discussion of diagnostic procedures and therapeutic strategies for HCC should occur in a multidisciplinary tumor board, including a liver transplant conference. The diagnostic algorithm for HCC is demonstrated in Figure 1.

## 3. Stage-Dependent Therapeutic Algorithm in HCC

The Barcelona Clinic Liver Cancer classification (BCLC classification) stands as the most widely utilized staging system for HCC (see Figure 2) [66]. This classification system takes into account not only tumor spread but also liver function and patient performance status, providing an estimation of overall survival prognosis based on tumor stage [67]. To evaluate performance status, the BCLC staging system employs the Eastern Cooperative Oncology Group (ECOG) scale [68]. Assessments of liver function can be stratified using the Model for End-Stage Liver Disease Score (MELD-Score), Child–Pugh score, and albumin–bilirubin score (ALBI-Score) [69,70,71,72]. Additionally, portal hypertension is a factor considered in decision-making for evaluating therapeutic strategies, especially in the assessment of tumor resection [67]. The current BCLC classification (update 2022) also incorporates AFP concentration as a prognostic marker [73,74]. The latest update of the BCLC classification in 2022 introduced stratification of the intermediate stage (BCLC Stage B) into three subcategories based on tumor burden and liver function [69,74]. The first subgroup comprises potential liver transplantation candidates who meet local extended criteria, considering factors such as tumor size, number of nodules, and AFP concentration [63,75,76]. The second subgroup includes patients not meeting the extended transplantation criteria but having preserved portal vein flow, adequate liver function, and well-defined tumor nodes, making them suitable candidates for transcatheter arterial chemoembolization (TACE). The third subgroup involves patients with extensive HCC liver involvement who remain asymptomatic with preserved liver function, who may benefit more from systemic therapy due to limited TACE effectiveness [69,74].

The concept of downstaging at BCLC stage B has emerged to assess patients for liver transplantation despite multilocular tumor localization. Downstaging involves a response to locoregional therapy, with tumor response as an indicator of tumor biological activity. While no general recommendation exists regarding the upper limit for downstaging, promising data exist for tumor response through neoadjuvant therapy aiming for inclusion in the MILAN criteria [77]. Patient outcomes with downstaging have been comparable to those undergoing primary transplantation within the MILAN criteria [78,79], primarily utilizing TACE [74].

The updated BCLC 2022 model (see Figure 2) recognizes the potential usefulness of transarterial radioembolization (TARE) for treating single tumors smaller than 8 cm (BCLC Stage 0/A) based on the results from the LEGACY trial. This is considered when other first-line therapies like resection, ablation, or transplantation are not feasible or have failed [80]. Additionally, TACE should be considered in BCLC Stage 0/A for patients not suitable for any first-line therapies or in case of failure of these approaches [74].

The novel concepts of treatment stage migration (TSM) and “untreatable progression” in the current BCLC classification define shifts in treatment recommendations or failures of chosen strategies. [74] TSM reflects a shift to an advanced stage despite initial characteristics suggesting a less advanced stage. In cases where first-line therapy is not feasible due to patient characteristics, recommendations involve considering the next most suitable option within the same stage or a treatment for a more advanced stage. “Untreatable progression” refers to TACE failure, recommending transitioning to the next stage of BCLC classification in such scenarios [69,74]. Repeated treatments with TACE often lead to continuously declining liver function, resulting in patients being unable to undergo systemic therapy after multiple TACE interventions. The concept of TSM allows a transition from TACE therapy to systemic therapy at a stage where the patient still has adequate liver function for oncological systemic therapy.

The concept of downstaging (conversion to TACE, local ablation, or liver transplantation after systemic therapy) in stage BCLC B staging is also addressed in clinical studies. In cases where liver transplantation is a potential treatment option, systemic therapy can be used to downstage the tumor while the patient awaits transplantation, increasing the likelihood of successful transplantation outcomes. However, currently, there is no clear recommendation on this matter in international guidelines.

Combinations (simultaneously or sequentially) of TACE and immunotherapy are actually under investigation in the DEMAND trial (NCT04224636) and EMERALD-1 trial (NCT03778957). The ongoing DEMAND study evaluates the safety and efficacy of atezolizumab plus bevacizumab prior to (sequentially as concept of downstaging) or in combination with TACE in patients with unresectable HCC. The EMERALD-1 trial evaluated patient outcomes with a combination of a systemic therapy (durvalumab plus bevacizumab, durvalumab monotherapy, or placebo) and TACE. Recently published data showed a significant progression-free survival (PFS) benefit for durvalumab/bevacizumab plus TACE compared with placebo plus TACE; for further details, see section below [81,82]. Despite this promising data, there is no recommendation in international guidelines.

In BCLC Stage C, the standard therapy remains oncologic systemic therapy (see Figure 3). However, with the approval of atezolizumab/bevacizumab and tremelimumab/durvalumab, the therapeutic landscape has undergone significant changes. Further details on these developments are discussed below.

**Figure 2 cancers-16-01831-f002:**
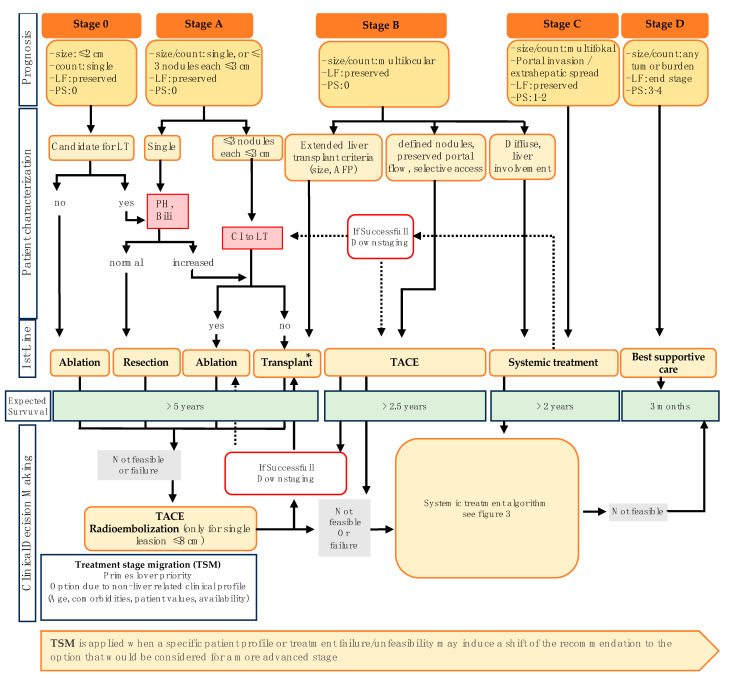
Modified Barcelona Clinic Liver Cancer (BCLC) classification as treatment algorithm for HCC (2022) [74]. * For bridging to transplant TACE, TARE and ablation are possible treatment options and are recommended by international guidelines [50,57,67,83]. Dashed line stands for treatment options currently being investigated in clinical trials but not currently recommended by guidelines. Bili: bilirubin; CI: contraindiacation; LF: liver function; LT: liver transplantation; PH: portal hypertension; PS: performance score; Stage 0: very early stage; Stage A: early stage; Stage B: intermediate stage; Stage C: advanced stage; Stage D: terminal stage; TACE: transarterial chemoembolization.

## 4. Current and Future Molecular Directed Therapeutic Agents in Systemic Treatment of HCC

### 4.1. Targeting Vascular Endothelial Growth Factor (VEGF) Signaling Pathway

The VEGF signaling pathway plays a crucial role for angiogenesis, which is the formation of new blood vessels. As a well-characterized mainstay of HCC tumorigenesis, this pathway becomes dysregulated, contributing to tumor growth and progression. The formation of new blood vessels from pre-existing vascular beds is a key characteristic of tumorigenic liver tissue [84]. Multitarget tyrosine kinase inhibitors (TKIs) that suppress angiogenesis by inhibiting VEGF receptors (VEGFRs) and platelet-derived growth factor receptors (PDGFRs) were among the first molecular directed therapies showing promising anti-tumor activity in advanced HCC. Additionally, TKIs influence tumor growth by inhibiting various downstream signaling cascades through the suppression of tyrosine kinases like Raf, c-kit, FGFR, and RET [85]. VEGF acts to stimulate the formation of new blood vessels from existing ones. In HCC, overexpression of VEGF and its receptors leads to increased tumor angiogenesis. This heightened blood vessel formation provides the tumor with nutrients and oxygen, facilitating its growth and enabling metastasis. As the VEGF pathway promotes specifically intratumoral angioneogenesis, disruption of VEGF signaling aims to restrict the blood supply to the tumor, thereby impeding its growth and potentially improving treatment outcomes for HCC patients.

Sorafenib marked the first approval as a multi-kinase inhibitor for treating advanced HCC, backed by data from the SHARP trial (NCT00105443). This phase III trial involved 602 patients with advanced HCC who had not undergone prior systemic treatment. Patients were randomized to receive sorafenib (at a dose of 400 mg twice daily) or a placebo. The sorafenib group demonstrated a median overall survival (mOS) of 10.7 months compared to 7.9 months in the placebo group. Additionally, the median time to radiologic progression was 5.5 months in the sorafenib group and 2.8 months in the placebo group [86]. Sorafenib primarily inhibits VEGFR, PDGFR, and the kinase Raf. By Raf kinase inhibition, it disrupts downstream signaling in the RAS/RAF/MEK/ERK pathway, leading to the suppression of cell division and proliferation. Inhibiting signal transduction at the VEGF receptor family (VEGFR1/2/3) and PDGFR restricts tumor angiogenesis and cell proliferation [87,88,89]. For a significant period, sorafenib has been the standard of care in advanced HCC and has served as the control group in various clinical trials. Ongoing research and a better understanding of side effects, along with more effective sequential therapies, have potentially contributed to a notable increase in the mOS data associated with sorafenib treatment (see Table 2).

Given the intricate pathogenesis of HCC, combining sorafenib with other molecular targeted drugs (such as MEK/ERK inhibitors, mTOR inhibitors, EGFR inhibitors, c-MET inhibitors) initially appeared promising and underwent investigation in several clinical trials. However, these combinations did not demonstrate a significant clinical benefit, as detailed below.

Several clinical trials also explored the combination of classic cytostatic chemotherapy with sorafenib for treating advanced HCC. In a phase II trial (INST 08-20, NCT01032850), 15 patients received capecitabine and sorafenib, resulting in a median overall survival (mOS) of 12.7 months (95% CI, 8.5–23.4) [90]. Another randomized phase II trial (ESLC01 study) compared tegafur–uracil plus sorafenib with sorafenib alone. Interestingly, the combination therapy showed no advantage in mOS (8.2 months for tegafur–uracil plus sorafenib vs. 10.5 months for sorafenib) [91]. The combination of sorafenib and gemcitabine/oxaliplatin was investigated in the PRODIGE 10 trial (phase II, NCT00941967) with 94 patients enrolled. They were randomized to gemcitabine/oxaliplatin plus sorafenib (arm A) and sorafenib alone (arm B). The mOS was 6.2 months (95% CI, 3.8–6.8) in arm A and 4.6 months (95% CI, 3.9–6.2) in arm B. The median progression-free survival (mPFS) was 13.5 months (95% CI, 7.5–16.2) in arm A and 14.8 months (95% CI, 12.2–22.2) in arm B. The overall response rate (ORR) was 15% in the gemcitabine/oxaliplatin plus sorafenib group and 9% in the sorafenib monotherapy group. While the study met its primary endpoint of achieving a 4-month PFS rate > 50%, it was not furthered to phase III due to the lack of significant benefit in mOS [92]. The combination of sorafenib with gemcitabine also fell short of achieving a clinically meaningful response in a single-arm phase II study. The study enrolled 45 patients, revealed an mPFS of 3.7 months (95% CI, 3.5–3.8), mOS of 11.6 months (95% CI, 7.4–15.9), and an ORR of 4% [93].

In summary, these trials indicate limited signals for an added benefit from sorafenib with cytotoxic chemotherapy. With the establishment of immunotherapy in treating advanced HCC, ongoing clinical trials are aimed at identifying which patient populations are more likely to benefit from TKI therapies including sorafenib and which cohort might profit from immunotherapy-based regimens (see Table 3).

Sorafenib was studied in combination with hepatic arterial infusion therapy (HAIC) for patients with advanced HCC and portal vein tumor thrombus (PVTT). A randomized phase III study (NCT02774187) demonstrated notable improvements in mPFS, mOS, and a higher response rate for sorafenib combined with HAIC of FOLFOX compared to sorafenib alone. Involving 247 patients, the combined treatment group exhibited an mOS of 13.37 months (95% CI, 10.27–16.46) compared to 7.13 months (95% CI, 6.28–7.98) in the sorafenib monotherapy group. The mPFS was 7.03 months (95% CI, 6.05–8.02) for HAIC/sorafenib versus 2.6 months (95% CI, 2.15–3.05) for sorafenib alone. The ORR significantly increased in the HAIC group at 51% compared to 3% in the sorafenib monotherapy group [94]. In another randomized phase III trial (NCT03009461), HAIC with the OFF protocol (oxaliplatin, 5-fluorouracil) in combination with sorafenib showed a significant improvement in survival, with an mOS of 16.3 months (95% CI, 0.0–35.5) compared to 6.5 months (95% CI, 4.4–8.6) with sorafenib alone [95]. However, the SCOOP-2 trial did not observe a survival benefit with the combination of HAIC (cisplatin) and sorafenib compared with sorafenib alone. Notably, the SCOOP-2 trial was underpowered for primary and secondary endpoints [96].

These studies were primarily conducted on an Asian patient population, where chronic hepatitis B drives HCC genesis and patients often present with good initial liver function. Further research is needed to ascertain the applicability of HAIC results to Western patient populations.

Moreover, ongoing studies are exploring new combination partners with HAIC, particularly involving immunotherapy approaches across neoadjuvant, adjuvant, and palliative settings (see Table 4). Recently data from a study combining HAIC, lenvatinib, and toripalimab for advanced HCC (NCT04044313) showed promising anti-tumor activity with an mPFS of 10.4 months, an mOS of 17.9 months, and an ORR of 63.9%) [97].

Lenvatinib became the second multi-kinase inhibitor approved by the U.S. Food and Drug Administration (FDA) in 2018 for treating advanced HCC. Apart from inhibiting VEGFR and PDGFR, lenvatinib also targets fibroblast growth factor receptor 1–4 (FGFR 1-4), c-kit, and RET, broadening its impact on associated signaling pathways involved in proliferation, cell differentiation, migration, angiogenesis, and apoptosis [98].

Approval of Lenvatinib was subjected from data in the REFLECT trial (phase III, NCT01761266), a non-inferiority study involving 954 randomized patients who received either lenvatinib (12 mg/day for bodyweight ≥ 60 kg or 8 mg/day for bodyweight < 60 kg) or sorafenib (400 mg twice daily). Lenvatinib demonstrated an mOS of 13.6 months (95% CI, 12.1–14.9) compared to the sorafenib group, which showed an mOS of 12.3 months (95% CI, 10.4–13.9). Moreover, mPFS increased from 3.7 to 7.4 months with lenvatinib. The study met its primary endpoint, proving non-inferiority of lenvatinib compared with sorafenib [99]. A recent subgroup analysis showcased an mOS of 21.6 months (95% CI, 18.6–24.5) for responders versus 11.9 months (95% CI, 10.7–12.8) for non-responders [100], emphasizing the significantly prolonged overall survival associated with tumor response to lenvatinib. It is important to note that the REFLECT study only included patients at Child–Pugh Stage A with preserved liver function. Ongoing studies for lenvatinib in HCC are listed in Table 5.

Clinical trials, including the REFLECT trial, have compared sorafenib and lenvatinib, used as first-line treatments for advanced HCC. These studies have shown comparable efficacy in terms of mOS and mPFS, and lenvatinib demonstrated non-inferiority to sorafenib in terms of mOS. Nevertheless, there is still no clear recommendation as to which TKI should preferably be used in first-line therapy if there are contraindications to immune checkpoint blockade. Various meta-analyses have been established with regard to this issue. A large meta-analysis including five clinical trials (one randomized clinical trial, four retrospective clinical trials) with 1481 patients showed no difference in overall survival and demonstrated a median overall survival of 13.4 months (95% CI; 9.38–17.48) in the lenvatinib group and 11.4 months (95% CI; 8.46–14.47) in the sorafenib group. Lenvatinib showed a significant improvement in PFS (HR 0.67, 95% CI; 0.48–0.94) with an mPFS of 5.88 months (95% CI; 3.68–8) compared to 4.17 months (95% CI; 3.08–5.25) in sorafenib patients, but this was not reflected in a survival benefit [101]. Another meta-analysis, including 15 studies with 3908 patients showed similar results with no significant difference in overall survival between lenvatinib and sorafenib (HR 0.86, 95% CI; 0.72–1.02) [102]. The safety profile with regard to higher-grade therapy-associated toxicity was comparable between the two groups. Patients on lenvatinib showed more hypertension, proteinuria, fatigue and weight-loss, whereas patients on sorafenib showed more frequent diarrhea and clinically significant hand–foot syndrome [102]. The decision for one of the two substances therefore depends primarily on the respective baseline patient characteristics, such as pre-existing arterial hypertension, skin alterations, or cachexia.

Regorafenib (inhibiting VEGFR, PDGFR, FGFR, RAF, RET, c-kit) and cabozantinib (inhibiting VEGFR, RET, KIT, MET, AXL), both multi-kinase inhibitors approved for advanced HCC treatment, are utilized as second-line treatment options following sorafenib failure [85]. Regorafenib is approved based on the results from the RESORCE trial (phase III, NCT01774344), where 573 were enrolled and randomized in a 2:1 manner to regorafenib (160 mg once daily) or placebo. Regorafenib exhibited a statistically significant extension in overall survival compared to the placebo group, with an mOS of 10.6 months (95% CI, 9.1–12.1) versus 7.8 months (95% CI, 6.3–8.8) [103]. However, the positions of regorafenib and also other treatment regimens after sorafenib failure warrant discussion, especially with the establishment of atezolizumab/bevacizumab as the new standard in first-line treatment based on IMbrave150 trial data.

Cabozantinib was assessed in the phase III CELESTIAL trial (NCT01908426) with 707 participants randomly assigned to cabozantinib (60 mg once daily) or placebo (2:1). Cabozantinib significantly improved overall survival compared to placebo, with an mOS of 10.2 months versus 8.0 months, respectively. mPFS also increased to 5.2 months with cabozantinib compared to 1.9 months with placebo [104]. The CELESTIAL study enrolled patients in both, second- and third-line treatment. Subgroup analysis highlighted the pronounced benefits of cabozantinib, particularly for second-line patients (mOS 11.3 months vs. 7.2 months in the overall second line cohort). Notably, the duration of prior sorafenib treatment did not impact overall survival [105].

Ongoing trials are investigating the impact of cabozantinib directly following therapy with immunocheckpoint inhibitors and in patients with impaired liver function (see Table 6).

Ramucirumab is a fully human recombinant IgG monoclonal antibody (mAb) targeting the VEGFR2 receptor that underwent investigation in the REACH trial (phase II, NCT01140347). This trial enrolled 565 patients with advanced HCC after sorafenib first-line therapy, randomized equally to receive either ramucirumab or a placebo. The primary endpoint was overall survival, resulting in an mOS of 9.2 months (95% CI, 8.0–10.6) for the ramucirumab group and 7.6 months (95% CI, 6.0–9.3) for the placebo group, without a significant difference between both groups [106]. However, a subgroup analysis revealed a survival benefit for patients with elevated AFP levels > 400 ng/mL.

Subsequently, the REACH-2 study (phase III, NCT02435433) confirmed the survival benefit of ramucirumab in patients with high AFP levels (>400 ng/mL). In this study, 292 patients were randomly assigned to receive either ramucirumab or placebo in a 2:1 ratio. The mOS was 8.5 months (95% CI, 7.0–10.6) in the ramucirumab group and 7.3 months (95% CI 5.4–9.1) in the placebo group, and mPFS was 2.8 months (95% CI, 2.8–4.1) and 1.6 months (95% CI, 1.5–2.7), respectively. Significantly improved mOS and mPFS were observed in the ramucirumab group [64]. Based on these findings, the FDA and EMA approved ramucirumab for the treatment of advanced HCC in patients with AFP levels > 400 ng/mL after prior sorafenib therapy.

Several other TKIs have been explored in clinical trials for treating advanced HCC. Brivanib (targeting VEGFR, FGFR), linifanib (targeting VEGFR, PDGFR), and sunitinib (targeting VEGFR, PDGFR, c-kit, RET) showed positive signals in phase II trials (see Table 7) [107,108,109], but these signals could not be substantiated in subsequent randomized phase III trials [110,111,112]. Tivozanib monotherapy demonstrated response and survival data akin to established TKIs in a phase I/II study. However, the study was terminated prematurely due to statistical shortcomings [113]. Cediranib, an oral pan-VEGFR inhibitor, yielded ineffective results in a phase II trial for patients with advanced HCC (mOS 5.8 months (95% CI, 3.4–7.3 months)) [114]. Dovotinib, an orally administered TKI, underwent a randomized phase II trial against sorafenib but failed to meet the primary endpoint of improved overall survival [115].

In 2021, a new multi-tyrosine kinase inhibitor, donafenib (targeting VEGFR, PDGFR, Raf kinase), demonstrated superiority over sorafenib in a phase II/III study (NCT02645981). The donafenib group showed significantly improved mOS (12.1 versus 10.3 months) [116]. Donafenib has gained approval in China for treating patients with unresectable hepatocellular carcinoma in a first-line setting, with a recommended dose of 200 mg twice daily. However, its potential value in Western patient populations remains uncertain, given that its approval in China was solely based on Chinese patient data. Notably, the control arm with sorafenib also exhibited relatively poor overall survival in the pivotal study for donafenib.

Apatinib, a novel tyrosine kinase inhibitor highly selective for VEGFR2, was assessed as a second-line treatment in advanced HCC within the randomized AHELP trial (phase III, NCT02329860). In this trial, 400 patients were randomly assigned to receive either apatinib (750 mg orally once daily) or placebo in a 2:1 ratio. The results revealed a significant improvement in overall survival within the apatinib-treated group compared to the placebo. The mOS was 8.7 months (95% CI, 7.5–9.8) in the apatinib group and 6.8 months in the placebo group (95% CI, 5.7–9.1) [117]. Due to these promising findings, apatinib is currently being investigated in various clinical trials as a potential targeted drug for combination therapies, such as those involving immune checkpoint inhibitors, TACE, and HAIC (as detailed in the section above). However, despite the encouraging data from the AHELP trial, apatinib has not yet received approval for the treatment of advanced HCC.

Anlotinib, an orally administered multitargeted tyrosine kinase inhibitor targeting VEGFR, FGFR, PDGFR, and c-kit, has shown promising anti-tumor efficacy in a phase II study (ALTER-0802, NCT02809534) evaluating its efficacy in first- or second-line treatment for patients with unresectable HCC [118]. The study comprised two cohorts: one with pre-treated patients and another with therapy-naive patients. The recommended dosage of anlotinib was 12 mg/day on days 1–14 of a 3-week cycle. The primary endpoint aimed for a 12-week PFS rate of 80% in cohort 1 (treatment naive) and 70% in cohort 2 (second-line after TKI). The achieved 12-week PFS rates were 80.8% and 72.5%, respectively. Moreover, the mOS was 12.8 months in cohort 1 and 18.0 months in cohort 2. Comparing overall survival is challenging owing to the variation in treatment lines. The authors reported a corrected mOS of 26.7 months in cohort 1, calculated from the commencement of first-line therapy to death. Meeting the primary endpoint, anlotinib is presently under continued investigation in numerous studies exploring its potential as a combination partner for immune checkpoint inhibition [119].

While addressing VEGF signaling can be effective in slowing down tumor growth, it also carries the risk of causing adverse effects or toxicity in patients. Some common toxicities associated with lenvatinib include hypertension, fatigue, weight loss, proteinuria, and hematotoxicity. Furthermore, gastrointestinal toxicity is often reported, including diarrhea, nausea, vomiting, gastrointestinal perforation or fistula and gastrointestinal bleeding, and elevated transaminases. Another therapy-limiting issue is the development of hand–foot syndrome. To reduce toxicity, a reduced starting dose of 8 mg/day is recommended for the use of lenvatinib in patients with a body weight of less than 60 kg. A comparison between patients in Study 202 and REFLECT reveals that efficacy was sustained while safety was enhanced in individuals with lower bodyweights who were administered lenvatinib at 8 mg/day in REFLECT, in contrast to those who received lenvatinib at 12 mg/day in Study 202 [120]. Another post hoc analysis of REFLECT data assessed lenvatinib efficacy and safety by body weight group. This retrospective analysis suggested that dosing lenvatinib based on body weight in patients with advanced HCC successfully upheld efficacy, with similar rates of treatment-emergent adverse events observed in both body weight groups [121]. There are also data for starting therapy with sorafenib and cabozantinib at a reduced dose, which showed a better side effect profile with preserved anti-tumor activity [122,123,124,125,126,127]. Specifically for hand–foot-syndrome, prophylactic application of emollients containing 10% urea, used two to three times daily, has been demonstrated to decrease the occurrence and postpone the onset of hand–foot syndrome in patients undergoing sorafenib treatment [128].

### 4.2. Immune Checkpoint Inhibitor (CPI) and Immune Checkpoint Inhibitor Combinations

So called immunocheckpoint receptors are expressed on the surface membrane of T lymphocytes that regulate and even diminish the intensity of the immune response to prevent overstimulation in a physiological state. In several solid tumors, immunocheckpoint ligands are upregulated, which bind to checkpoint receptors on T lymphocytes, allowing tumor cells to evade anti-tumor immune responses by being tolerated by the immune system. Inhibitory immunocheckpoints, which can be targeted by drugs, include the cytotoxic T-lymphocyte-associated Protein 4 (CTLA-4), programmed cell death protein 1 (PD-1), lymphocyte-activation gene 3 (LAG3), T-cell immunoglobulin and mucin domain containing-3 (TIM-3) and T-cell immunoreceptor with Ig and ITIM domains (TIGIT) [129].

Immunocheckpoint inhibitors are monoclonal antibodies that block the interaction between immunocheckpoint receptors and their ligands. By doing so, they enhance the body’s anti-tumor immune response, primarily by activating T cells. The ligands of PD-1 are programmed cell death ligand 1 (PD-L1) and programmed cell death ligand 2 (PD-L2), while CTLA-4 binds to cluster of differentiation 80/86 (CD80/86). PD-L1 and PD-L2 expression occurs in various tumor cells, including HCC, transmitting inhibitory signals to T cells and promoting immune escape mechanisms in tumor cells.

In HCC, CPIs primarily target PD-1, PD-L1, and cytotoxic CTLA-4. HCC often creates an immunosuppressive microenvironment, allowing cancer cells to evade immune detection and attack. CPIs work by blocking these checkpoint proteins, reactivating the immune system’s ability to recognize and eliminate cancer cells. The use of CPI monotherapy in HCC has previously shown promising results in some patients, leading to durable responses and improved survival rates. However, not all patients respond equally to these therapies, and the effectiveness can vary based on factors like the tumor microenvironment, level of immune cell infiltration, and the expression of specific biomarkers like PD-L1. Combination therapies involving CPIs with other treatment agents or modalities, such as targeted therapies or locoregional treatments like transarterial chemoembolization (TACE) or radiofrequency ablation (RFA) are also being explored. These combinations aim to enhance the overall anti-cancer immune response and potentially improve treatment outcomes for patients with HCC (for further details, see section below).

CheckMate 040 was a large phase I/II multi-cohort trial (NCT01658878) assessing safety and efficacy of nivolumab (PD-1 inhibitor) in patients with advanced HCC previously treated with sorafenib. The trial included a phase 1 dose-finding protocol (cohort 1) with 48 patients (*n* = 9 sorafenib-naive, *n* = 37 sorafenib-pretreated), leading to the selection of a 3 mg/kg dosage for the subsequent dose expansion phase [130]. In the phase Ib dose expansion protocol (cohort 2) involving 214 patients (*n* = 69 sorafenib-naive, *n* = 145 sorafenib-pretreated) across four arms (sorafenib-untreated or intolerant without viral hepatitis, sorafenib progressor without viral hepatitis, HCV-infected, and HBV-infected), nivolumab monotherapy was administered. The ORR was 20% (95% CI, 15–26) in the dose expansion part and 15% (95% CI, 6–28) in the dose escalation part. For patients previously treated with sorafenib in the dose escalation phase, the mOS was 15 months (95% CI, 9.6–20.2) [130]. In the final report of patients with advanced HCC who progressed under sorafenib therapy or were intolerant to sorafenib and received nivolumab 3 mg/kg in the dose escalation or dose expansion phase (*n* = 154), the ORR was 14.3% (95% CI, 9.2–14.3), the mOS was 15.15 months (95% CI, 13.24–18.14) and the mPFS was 2.83 months (95% CI, 2.66–4.04). Based on these findings, in 2017, the FDA granted accelerated approval for nivolumab in treating HCC in patients previously treated with sorafenib [131]. In 2021, approval for nivolumab was revoked because of the negative outcomes of the CheckMate 459 trial, which examined the efficacy of nivolumab in the first-line setting. This phase III trial (NCT02576509) compared nivolumab versus sorafenib in the first-line setting, and 1009 patients were enrolled and randomly assigned to receive nivolumab (240 mg intravenously every 2 weeks) or sorafenib (400 mg orally twice daily). Despite enrolling 1009 patients, the trial did not achieve statistical significance for the primary endpoint of overall survival. The mOS was 16.4 months (95% CI, 13.9–18.4) in the nivolumab group and 14.7 months (95% CI, 11.9–17.2) in the sorafenib group (hazard ratio 0.85 (95% CI, 0.72–1.02); *p* = 0.075) [132].

Cohort 4 of phase I/II CheckMate 040 studied nivolumab in combination with ipilimumab in patients with advanced HCC in the second-line setting. Safety and efficacy were explored across three dosing arms. Among these, arm A—comprising ipilimumab 3 mg/kg plus nivolumab 1 mg/kg every 3 weeks for four doses, followed by nivolumab monotherapy every 2 weeks (240 mg)—demonstrated the most promising outcomes. Arm A reported an ORR of 32% irrespective of baseline etiology or PD-L1 status, with an mOS of 22.8 months (95% CI, 9.4—not reached) [133]. Consequently, in 2021, the FDA granted accelerated approval in the US for nivolumab plus ipilimumab to treat HCC in patients previously treated with sorafenib. Cohort 5 of CheckMate 040 evaluated nivolumab plus ipilimumab in patients with Child–Pugh B cirrhosis and showed favorable safety with manageable toxicities [134].

The active CheckMate 9DW (NCT04039607) phase III trial is currently evaluating the combination of ipilimumab plus nivolumab against sorafenib or lenvatinib in first-line therapy in a palliative setting. A press-release announced that the CheckMate 9DW trial met its primary endpoint of overall survival for the first-line treatment of advanced HCC.

Pembrolizumab, a monoclonal antibody targeting the PD-1 receptor, was assessed in patients with advanced HCC previously treated with sorafenib in the KEYNOTE-224 trial (phase II, NCT02702414). This study enrolled 104 patients who received pembrolizumab (200 mg) every 3 weeks. Pembrolizumab demonstrated promising clinical efficacy with a reported ORR of 17% (95% CI, 11–26), an mOS of 12.9 months (95% CI, 9.7–15.5), and an mPFS of 4.9 months (95% CI, 3.4–7.2) [135]. Updated efficacy results were reported in 2022, reflecting an extended follow-up of 2.5 years, revealing an ORR of 18.3% (95% CI, 11.4–27.1), an mPFS of 4.9 months (95% CI; 3.5–6.7), and an mOS of 13.2 months (95% CI, 9.7–15.3) [136]. Consequently, in 2018, based on these findings, the FDA granted accelerated approval to pembrolizumab for the treatment of advanced HCC in patients previously treated with sorafenib.

The subsequent randomized KEYNOTE-240 trial (phase III, NCT02702401) compared pembrolizumab versus placebo in the second-line setting for patients with advanced HCC. While the reported results were similar to those in the KEYNOTE-224 trial, statistical significance for OS and PFS was not achieved according to specified criteria [137]. A recently published long-term analysis, with a follow-up of approximately 40 months, revealed an mOS of 13.9 months (95% CI, 11.6–16.0) for pembrolizumab versus 10.6 months (95% CI, 8.3–13.5) for placebo. The mPFS was 3.3 months (95% CI, 2.8–4.1) for pembrolizumab versus 2.8 months (95% CI, 1.6–3.0) for placebo, while the ORR was 18.3% for pembrolizumab and 4.4% for placebo [138]. Despite not meeting prespecified statistical significance, pembrolizumab continued to demonstrate promising anti-tumor activity especially in patients that initially responded to therapy. In the KEYNOTE-394 trial (phase III, NCT03062358), pembrolizumab versus best supportive care was investigated in an Asian population with previously treated advanced HCC. A total of 453 patients were enrolled, and the primary endpoint was overall survival. The study met its primary endpoint, demonstrating an mOS of 14.6 months (95% CI, 12.6–18.0) in the pembrolizumab group compared to 13.0 months (95% CI, 10.5–15.1) in the best supportive care group. Pembrolizumab also statistically significantly improved secondary endpoints, displaying an mPFS of 2.6 months (95% CI, 1.5–2.8) and an ORR of 12.7% (95% CI, 9.1–17.0). In contrast, the best supportive care arm showed an mPFS of 2.3 months (95% CI, 1.4–2.8) and an ORR of 1.3% (95% CI, 0.2–4.6) [139].

The PD-1 inhibitor tislelizumab was investigated in the RATIONALE-208 trial (phase II, NCT03419897) among patients with previously treated advanced HCC. A total of 249 patients were enrolled and received tislelizumab monotherapy (200 mg, intravenously) every 3 weeks. The observed ORR was 13% (95% CI, 9–18), and this response was independent of the number of prior therapies. Moreover, the mOS was 13.2 months (95% CI, 10.8–15.2) [140]. Given the promising results from RATIONALE-208, a subsequent randomized phase III trial, RATIONALE-301 (NCT03412773), was initiated to confirm the efficacy of tislelizumab in first-line treatment of advanced HCC. In this trial, 674 patients were randomized to receive either tislelizumab or sorafenib. The primary study endpoint aimed to establish non-inferiority compared to sorafenib in terms of median overall survival. The study successfully met its primary endpoint, revealing an mOS of 15.9 months in the tislelizumab group compared to 14.1 months in the sorafenib group. Notably, the ORR was 14.3% for patients receiving tislelizumab versus 5.4% in the control arm, while the median PFS was 2.2 months for tislelizumab and 3.6 months for sorafenib [141]. Despite the encouraging results observed in the phase III trial, tislelizumab has not received FDA or EMA approval at present.

Camrelizumab, another PD-1 inhibitor, underwent evaluation in a phase II study (NCT02989922) involving patients with advanced HCC who had previously failed at least one systemic treatment line. In this study, 217 patients were randomly assigned to receive camrelizumab at a dose of 3 mg/kg (intravenously) every 2 or 3 weeks. The ORR was 14.7% (95% CI, 10.3–20.2) and the mOS was 13.8 months (95% CI, 11.5–16.6). The study provided evidence of anti-tumor efficacy. Presently, there are no available phase III studies for camrelizumab monotherapy. However, a recent investigation into its combination with apatinib is discussed further below.

Following the promising data from CheckMate 040 on the efficacy of combining nivolumab and ipilimumab, the approach of dual immune-checkpoint blockade in HCC therapy is currently under investigation in clinical trials. The combination of durvalumab (PD-L1 inhibitor) and tremelimumab (CTLA-4 inhibitor) were examined in the HIMALAYA trial. This phase III trial (NCT03298451) enrolled 1171 patients who were randomly assigned to three groups: patients receiving durvalumab (1500 mg every 4 weeks, intravenously) along with a single dose tremelimumab (300 mg, intravenously), durvalumab monotherapy, or sorafenib monotherapy. The mOS was 16.43 months (95% CI, 14.16–19.58) for the combination of durvalumab and tremelimumab, 16.56 months (95% CI, 14.06–19.12) for patients treated with durvalumab alone and 13.77 months (95% CI, 12.25–16.13) in the sorafenib treatment group. The combination of durvalumab and tremelimumab demonstrated superiority over sorafenib therapy. Furthermore, monotherapy with durvalumab showed non-inferiority compared to sorafenib therapy [142]. Based on data from the HIMALAYA trial, the FDA and EMA approved the combination of durvalumab plus tremelimumab for the treatment of advanced HCC in therapy-naive patients.

However, there is currently no approval for using durvalumab as monotherapy. In a phase II clinical trial (NCT01008358), tremelimumab monotherapy demonstrated favorable anti-tumor efficacy in patients with advanced HCC developed on HCV-induced liver cirrhosis. Durvalumab and tremelimumab as single agents compared with the combination of durvalumab and tremelimumab has been further investigated in a phase I/II trial (NCT02519348). The recommended dosage for tremelimumab and durvalumab, known as the STRIDE protocol, was also investigated in this trial and showed the most encouraging efficacy and safety profile. The STRIDE protocol demonstrated an ORR of 24.0% (95% CI, 14.9–35.3) and an mOS of 18.7 months (95% CI, 10.8–27.3). Durvalumab monotherapy showed an ORR of 10.6% (95% CI, 5.4–18.1) and an mOS of 13.6 months (95% CI, 8.7–17.6), while tremelimumab monotherapy showed an ORR of 7.2% (95% CI, 2.4–16.1) and an mOS of 15.1 months (95% CI, 11.3–20.5) [143].

Recently, there has been substantial exploration into targeting the immune checkpoints LAG3 and TIGIT through various clinical trials. Prolonged exposure to antigens stimulates a state of dysfunctional and exhausted T cells, subsequently leading to the upregulation of immune checkpoints such as PD-1, LAG3, and TIGIT. In the case of cancer, the ligands associated with these immune checkpoints are expressed by both tumor cells and antigen-presenting cells (APCs) within the tumor microenvironment. Under normal conditions, LAG3 remains absent in naive T cells. However, upon stimulation of CD4+ and CD8+ T cells, upregulation of LAG3 occurs. This upregulation serves to moderate the magnitude of immune response and prevent autoimmune processes. LAG3 is a negative regulator of proliferation, activation, and homeostasis in T cells. One of its primary ligands, fibrinogen-like protein 1 (FGL1), is notably upregulated on the cell surfaces of solid tumor cells [144,145]. In cancer patients, TIGIT is also upregulated by activated T cells, natural killer cells, and regulatory T cells. CD155 and CD112 serve as important ligands for TIGIT and expressed by both tumor cells and antigen-presenting cells in the tumor microenvironment [146].

Relatlimab, an LAG3 inhibitor, in combination with nivolumab is currently under investigation in the RELATIVITY-073 trial (phase II, NCT04567615). The trial enrolled patients with advanced HCC who are immunotherapy-naive but showed tumor progress while undergoing therapy with TKI. The eagerly anticipated results of this trial are pending. Additionally, there is ongoing research on the combination of relatlimab with nivolumab and bevacizumab in the RELATIVITY-106 trial (phase III, NCT05337137).

In a phase I/II study (NCT02460224) evaluating the LAG-3 inhibitor ieramilimab in patients with advanced solid tumors, including seven patients with HCC, promisingly, two out of seven patients with HCC showed a stable disease as best response [147].

The recently published MORPHEUS-Liver study (phase Ib/II, NCT04524871) explored the combination therapy with tiragolumab, a TIGIT inhibitor, in patients with advanced HCC. In this study, 58 patients were enrolled and randomly assigned in a 2:1 ratio to receive either tiragolumab (300 mg intravenously every 3 weeks) combined with atezolizumab and bevacizumab or the standard treatment with atezolizumab and bevacizumab alone. The trial demonstrated a notable increase in ORR. The tiragolumab group reached an impressive ORR of 42.5% compared with 11.1% in the control group. The mPFS significantly differed between the groups, with 11.1 months for the tiragolumab triple combination versus 4.2 months for the control group. Interestingly, there were no significant differences observed in ORR and PFS between the subgroups based on PD-L1 status [148]. Furthermore, it is crucial to note the poor performance of the control group, showing an ORR of 11.5% under atezolizumab and bevacizumab, is unequivocal below the expected data from the IMbrave-150 study (ORR 27.3%) [149]. Another phase II trial (AdvanTig206, NCT04948697) investigated the triple combination of tislelizumab plus BAT1706 (bevacizumab biosimilar) and ociperlimab (anti-TIGIT mAb) against tislelizumab plus BAT1706. Initial data presented at ESMO 2023 exhibited an ORR of 35.5% for the triplet therapy and 37.5% for doublet therapy, without demonstrating a statistically significant difference. Similarly, the mPFS did not significantly vary between the groups (8.3 months versus 6.9 months). Furthermore, a subgroup analysis indicated a potential advantage in terms of ORR for tumors with PD-L1 status greater than 1.

There are currently numerous clinical studies on the use of immune checkpoint inhibitors for the treatment of HCC (see Table 8).

Immune-related adverse events (irAE) are among the main side effects of checkpoint inhibitors and represent a therapy-limiting challenge. In the HIMALAYA trial, grade 3 or 4 immune-mediated events were observed in 49 patients (12.6%) [142]. A meta-analysis of 47 studies including 6472 patients reported an all-grade irAE incidence rate of 34% (95% CI; 22–47%) and the rate of events ≥ grade 3 was 9% (95% CI; 5–14%) [150]. The most observed irAEs affect the skin, colon, endocrine organs, liver, and lungs. While others are rare, they can be severe, even life-threatening, including neurological disorders and myocarditis. In the context of treating patients with HCC who are receiving therapy with immune checkpoint inhibitors, regular monitoring for the occurrence of immune-mediated adverse events should be ensured. Generally, upon reaching a grade 2 toxicity, treatment interruption and initiation of corticosteroid therapy are recommended. Detailed management recommendations for specific immune therapy-mediated affected organs are outlined in international guidelines [151,152]. Interestingly, the occurrence of irAEs was linked to improved survival outcomes in real-world cohorts of patients with advanced HCC receiving treatment with atezolizumab plus bevacizumab [153,154].

### 4.3. Combined Inhibition of VEGF Signaling Pathway and CPI

The simultaneous blockade of VEGF signaling and immunocheckpoints demonstrates a broad pathophysiological foundation, eliciting synergistic or at least additional effects on tumor growth and transformation of the tumor microenvironment from an immunosuppressive to an immunostimulatory state. VEGFs lead not only to tumor growth and increased metastatic potential through increased angiogenesis but also to stimulation of T-cell invasion and release of immunosuppressive cytokines in the tumor microenvironment [155]. Anti-tumor efficacy for treatment strategies targeting VEGF signaling and immunocheckpoints were reported in several studies (see Table 9).

The combination of bevacizumab (a VEGF inhibitor) and atezolizumab (a PD-L1 inhibitor) marked the first extensively studied combination for therapy of advanced HCC. Bevacizumab, either as monotherapy or combined with cytotoxic chemotherapy, has demonstrated meaningful anti-tumor activity and a manageable safety profile in various phase II studies [156,157,158]. The investigation of bevacizumab and atezolizumab commenced with a phase Ib trial (GO30140, NCT02715531) involving patients with advanced HCC and without previous systemic treatment. Cohort A, consisting of 104 patients, received treatment with atezolizumab (1500 mg) and bevacizumab (15 mg/kg) every 3 weeks. The ORR was 36% (95% CI; 26–46). In cohort F, 119 patients were randomly assigned to receive atezolizumab plus bevacizumab or atezolizumab alone. The combination therapy displayed a longer mPFS of 5.6 months compared to 3.4 months in the atezolizumab monotherapy group [159]. Based on these positive results of the combination therapy, a randomized phase III trial (IMbrave150, NCT03434379) was initiated. This study enrolled 501 patients with advanced HCC and without previous systemic treatment. Patients were randomly assigned in a 2:1 ratio to receive either atezolizumab (1500 mg) and bevacizumab (15 mg/kg) every 3 weeks or sorafenib (400 mg twice daily). Atezolizumab plus bevacizumab demonstrated statistically significant superiority over sorafenib in terms of both mOS and mPFS [149]. The updated efficacy data revealed an mOS of 19.2 months (95% CI; 17.0–23.7) for patients treated with atezolizumab plus bevacizumab, compared to 13.4 months (95% CI; 11.4–16.9) with sorafenib. The mPFS was 6.9 months (95% CI; 5.7–8.6) in the atezolizumab plus bevacizumab group and 4.3 months (95% CI; 4.0–5.6) in the control group. Additionally, the ORR was reported to be 30% for patients treated with atezolizumab plus bevacizumab [160]. These compelling results led to the FDA and EMA approvals for the combination of atezolizumab and bevacizumab for the treatment of advanced HCC as a first-line treatment. However, the optimal sequence of therapy after atezolizumab/bevacizumab remains uncertain due to the lack of randomized trial data on second-line treatment following atezolizumab/bevacizumab. The ongoing IMbrave251 trial (phase III, NCT04770896) aims to address this gap by comparing atezolizumab plus lenvatinib or sorafenib versus lenvatinib or sorafenib therapy alone in patients who progressed after prior systemic treatment with atezolizumab plus bevacizumab. Furthermore, it is actually unknown whether atezolizumab/bevacizumab is more efficacious than TACE for the treatment of intermediate stage HCC. The ABC-HCC trial (phase III, NCT04803994) randomized patients with intermediate stage HCC (1:1) atezolizumab/bevacizumab or TACE [161].

The use of atezolizumab/bevacizumab was tested against sorafenib, the first-line standard at the time, in the IMbrave150 trial. Results from randomized controlled clinical phase III trials for atezolizumab/bevacizumab against lenvatinib are not available. A retrospective multicenter study compared atezolizumab/bevacizumab versus lenvatinib in a first-line setting. The retrospective analysis involved 1341 patients treated with lenvatinib and 864 patients treated with atezolizumab plus bevacizumab. Atezolizumab plus bevacizumab did not demonstrate a survival advantage compared to lenvatinib (HR 0.97; *p* = 0.739). However, in patients with viral etiology, atezolizumab plus bevacizumab showed a significant extension in overall survival (HR 0.76; *p* = 0.024). Conversely, lenvatinib prolonged overall survival in patients with non-alcoholic steatohepatitis (HR 1.88; *p* = 0.014). Atezolizumab plus bevacizumab demonstrated a superior safety profile for most recorded adverse events [162]. Furthermore, the pivotal study of atezolizumab/bevacizumab only included patients with Child–Pugh A liver function, so the efficacy in patients with impaired liver function is unclear. Another retrospective analysis investigated the use of atezolizumab/bevacizumab versus lenvatinib in patients with Child–Pugh B status and advanced BCLC B or C stage HCC. In total, 217 patients with Child–Pugh B status were enrolled (30% received atezolizumab plus bevacizumab, while 70% received lenvatinib). Patients treated with lenvatinib demonstrated an mOS of 13.8 months (95% CI; 11.6–16.0), whereas those receiving atezolizumab plus bevacizumab demonstrated an mOS of 8.2 months (95% CI; 6.3–10.2). Atezolizumab plus bevacizumab showed inferior overall survival compared to lenvatinib (HR 1.9, 95% CI; 1.2–3.0, *p* = 0.0050), with no significant differences in mPFS. Among patients treated with atezolizumab plus bevacizumab, those with Child–Pugh B status, ECOG PS 0, BCLC B stage, or ALBI grade 1 derived similar survival benefits as those receiving lenvatinib [163]. The interpretation of data from retrospective studies remains challenging, so ultimately only prospective, randomized clinical trials can provide information on the value of lenvatinib versus atezolizumab/bevaicuzumab in first-line therapy.

The therapy with atezolizumab and bevacizumab showed an overall tolerable safety profile. In addition to irAEs due to atezolizumab, the main bevacizumab-associated side effects were arterial hypertension and proteinuria. Furthermore, the IMbrave150 pivotal study showed a high incidence of bleeding events, including from esophageal varices [149]. Updated efficacy and safety data from IMbrave150 reported an incidence of proteinuria in 29% and hypertension in 28% of patients receiving atezolizumab/bevacizumab [160]. The IMbrave150 study notably excluded patients with untreated or incompletely treated varices and bleeding or those at high risk of bleeding. The IMbrave150 trial reported esophageal variceal bleeding in 2.4% of patients treated with atezolizumab/bevacizumab and 0.6% of patients treated with sorafenib [149]. A retrospective analysis of real-world data treatment with atezolizumab/bevacizumab demonstrated a higher incidence of gastrointestinal bleeding events (14%) [164]. Before starting treatment with atezolizumab/bevacizumab, the esophageal variceal status should be examined. Endoscopic monitoring throughout treatment is recommended, and adherence to primary and secondary prophylaxis of variceal bleeding is essential [165]. For patients with esophageal varices or existing portal hypertension, an alternative therapy with durvlaumab/tremelimumab is also available. No evidence of an increased incidence of esophageal variceal bleeding has been reported to date for dual immune-checkpoint blockade [165].

Pembrolizumab in combination with lenvatinib was initially investigated in a phase Ib study (KEYNOTE-524, NCT03006926). This study enrolled 104 treatment-naive patients who received pembrolizumab (200 mg) every 3 weeks and lenvatinib daily (12 mg for bodyweight ≥ 60 kg; 8 mg bodyweight < 60 kg). The ORR and mPFS, assessed by mRECIST, were recorded as 36.0% and 8.6 months, respectively. The combination of lenvatinib plus pembrolizumab exhibited promising anti-tumor efficacy, revealing an mOS of 22 months with a manageable safety profile [166]. In 2019, the FDA granted accelerated approval for lenvatinib plus pembrolizumab in patients with advanced HCC and no previous systemic treatment. However, following emerging data on atezolizumab plus bevacizumab, the FDA withdrew accelerated approval for lenvatinib plus pembrolizumab in 2020.

Recently, results from the LEAP-002 trial (phase III, NCT03713593) were published, aiming to evaluate the safety and efficacy of lenvatinib plus pembrolizumab versus lenvatinib monotherapy in a first-line setting for patients with advanced HCC. The mOS with lenvatinib plus pembrolizumab was 21.2 months (95% CI; 19–23.6), while with lenvatinib alone it was 19.0 months (95% CI; 17.2–21.7). The mPFS and ORR were 8.2 months (95% CI; 6.3–8.3) and 26.1% (95% CI; 21.8–30.7) for pembrolizumab plus lenvatinib and 8.1 months (95% CI; 6.3–8.3) and 17.5% (95% CI; 13.9–21.6) for lenvatinib monotherapy. However, the LEAP-002 trial, with the dual primary endpoint of OS and PFS, did not meet the prespecified statistical significance [167]. Notably, tumor response measured by mRECIST1.1 appeared more favorable in the combination treatment group. Additionally, an issue encountered in the trial was the above-average performance observed in the control arm.

The combination of nivolumab with lenvatinib underwent evaluation in a phase Ib trial (NCT03418922), demonstrating an ORR assessed by mRECIST of 76.7% [168]. In the IMMUNIB trial (phase II, NCT03841201), lenvatinib plus nivolumab was investigated as a first-line treatment for patients with advanced HCC. Despite achieving an ORR of 28% the trial failed to reach the prespecified ORR target of at least 40%. Nevertheless, the trial revealed promising anti-tumor efficacy. IMMUNIB demonstrated an mOS of 27.1 months and an mPFS of 9 months, suggesting notable efficacy of nivolumab plus lenvatinib in treating patients with advanced HCC [169].

The RESCUE trial (phase II, NCT03463876) investigated the efficacy and safety of apatinib (250 mg once daily) combined with camrelizumab (200 mg for bodyweight ≥ 50 kg; 5 mg/kg for bodyweight < 50 kg) for treating patients with advanced HCC. In total, 70 patients were enrolled, including treatment-naive individuals or those with prior systemic treatment. The mPFS was 5.7 months (95% CI; 5.4–7.4) for treatment-naive patients and 5.5 months (95% CI; 3.7–5.6) for previously treated patients. The ORR was 34.3% in the first-line cohort and 22.5% in the second-line cohort. Impressively, the combination with camrelizumab and apatinib demonstrated high histopathologic response rates in a neoadjuvant setting during a phase II/III study (NCT04521153), and 42 out of 60 patients who underwent surgery exhibited a major pathologic response after neoadjuvant treatment with camrelizumab and apatinib [170]. Continuing this exploration, the CARES-310 trial (phase III, NCT03764293) further evaluated camrelizumab in combination with apatinib. In total, 543 patients with advanced HCC and without prior systemic treatment were enrolled and randomly assigned to receive camrelizumab plus apatinib or sorafenib monotherapy. The combination group displayed a significantly increased mPFS of 5.6 months (95% CI; 5.5–6.3) compared to the control group with an mPFS of 3.7 months (95% CI; 2.8–3.7). Similarly, the mOS was notably prolonged with the combination treatment (22.1 months; 95% CI; 19.1–27.2) compared to sorafenib treatment (15.2 months; 95% CI; 13.0–18.5) [171]. Based on the impressive survival data, the combination of apatinib and camrelizumab represents a potential therapeutic option in the first-line treatment for patients with advanced HCC. Notably, this combination has gained approval only in China. Currently, the combination of apatinib and camrelizumab lacks FDA or EMA approval. However, FDA approval is anticipated by May 2024, considering the current clinical progression.

The COSMIC-312 trial (phase III, NCT03755791) investigated the efficacy of cabozantinib plus atezolizumab in first-line treatment for patients with advanced HCC. Among the 432 patients treated with atezolizumab in combination with cabozantinib and the 271 patients treated with sorafenib as active comparator, the combination treatment group demonstrated an mOS of 15.4 months (96% CI; 13.7–17.7) and an mPFS of 6.8 months (99% CI; 5.6–8.3). In the control group, receiving sorafenib, the mOS was 15.5 months, and the mPFS was 4.2 months [172]. The trial successfully met one of its primary endpoints by revealing a significant improvement in PFS with cabozantinib plus atezolizumab compared to sorafenib in the first-line setting. Nonetheless, cabozantinib plus atezolizumab did not lead to an improvement in overall survival. Consequently, the sponsor recently announced the decision not to submit an application to the FDA for approval of this combination. Notably, the control arm, with an impressive performance with an mOS of 15.5 months for sorafenib, is noteworthy and stands out as unprecedented in previous observations.

Further evidence supporting the efficacy of cabozantinib in combination with checkpoint inhibitors in the second-line treatment of advanced HCC is available from cohort 6 of the CheckMate 040 trial (NCT01658878). In this cohort, 71 patients were enrolled and randomly assigned to receive treatment with cabozantinib plus ipilimumab and nivolumab or ipilimumab plus nivolumab. The ORR was 17% (95% CI; 6–33) in the group treated with cabozantinib plus ipilimumab and nivolumab and 29% (95% CI; 15–46) in the group receiving only dual checkpoint inhibition (ipilimumab and nivolumab). The mPFS was 5.1 months, and the mOS was 20.2 months in the triplet arm. Meanwhile, the doublet arm showed an mPFS of 4.3 months and an mOS of 22.1 months [173].

The efficacy of sintilimab in combination with the bevacizumab-biosimilar IBI305 in patients with systemic treatment-naive advanced HCC was investigated in the ORIENT-32 trial (phase II/III, NCT03794440). Patients were randomly assigned to receive either sintilimab plus IBI305 or sorafenib. In the combination treatment group, the mPFS was 4.6 months (95% CI; 4.1–5.7), notably higher than the control group with an mPFS of 2.8 months (95% CI; 2.7–3.2). Sintilimab plus IBI305 also demonstrated a significant improvement for overall survival. The ORR was 25.1% for sintilimab plus IBI305 and 7.7% for sorafenib [174,175]. Based on these data, the combination of sintilimab and IBI305 has received approval in China. Ongoing studies for combination of target VEGF and CPIs in HCC are reported in Table 10.

**Table 9 cancers-16-01831-t009:** Results from phase I/II studies for combination of target VEGF and CPIs in HCC.

Study	Phase	Setting	Treatment	ORR	mPFS	mOS	*n*	NCT
RESCUE	II	1L, palliative 2L, palliative	Apa + Cam	34.4%; 22.5%	5.7 months; 5.5 months	-	70	NCT03463876
-	I	palliative	Nivo + Len	76.7%	-	-	30	NCT03418922
IMMUNIB	II	1L, palliative	Nivo + Len	28%	9 months	27.1 months	50	NCT03841201
KEYNOTE-524	I	1L, palliative	Pembro + Len	36%	8.6 months	22.0 months	104	NCT03006926
GO30140	I	1L, palliative	A + B	36%	-	-	104	NCT02715531
ALTER-H003	II	1L, palliative	Anlo + Tori	32.3%	11.0 months	18.2 months	31	-
KEEP-G04	II	1L, palliative	Anlo + Sinti	55.0%	12.2 months	-	20	NCT04052152
VEGF Liver 100	I	1L, palliative	Ave + Axi	31.8%	-	-	22	NCT03289533
KEYNOTE-743	I	1L, palliative	Pembro + Rego	31%	7.5 months	26.5 months	35	NCT03347292
RENOBATE	II	1L, palliative	Nivo + Rego	35.7%	7.4 months	not reached	42	NCT04310709
-	II	1L, palliative	Tori + B	46.2%	9.9 months	not reached	54	NCT04605796
JVDJ	I	2L, palliative	Dur + Ramu	11%	4.4 months	10.7 months	28	NCT02572687
CAMILLA	I	2L, palliative	Cabo + Dur	66.6%	-	-	3	NCT03539822
DEDUCTIVE	I/II	1L, palliative	Tivo + Dur	28.6%	-	-	7	NCT03970616
-	II	neoadjuvant	Sinti + Len	36.1%	-	-	36	NCT04042805
-	-	1L, palliative	Len + Cam	41.7%	10.3 months	not reached	92	[176]
-	II	1L, palliative	Sor + Tori	35.7%	4.8 months	-	28	NCT04926532

1L: first-line; 2L: second-line; A: atezolizumab; Anlo: anlotinib; Apa: apatinib; Ave: avelumab; Axi: axitinib; B: bevacizumab; Cabo: cabozantinib; Cam: camrelizumab; Dur: durvalumab; Len: lenvatinib; mPFS: median progression-free survival; mOS: median overall survival; *n*: number of enrolled patients; ORR: overall response rate; Pembro: pembrolizumab; Ramu: ramucirumab; Rego: regorafenib; Sinti: sintilimab; Sor: sorafenib; Tivo: tivozanib; Tori: toripalimab.

### 4.4. Combination VEGF Signaling Pathway and/or CPI with TACE

The combination of systemic therapy with a localized procedure like TACE appears to be beneficial for patients with locally advanced tumors that are no longer suitable for resection or ablation. TACE is generally recommended for patients with HCC classified as stage BCLC B. The revised BCLC classification has repositioned systemic therapy, previously directed mainly to patients in stage BCLC C, towards those already in stage BCLC B (termed treatment stage migration). Also, the concept of tumor downstaging has repositioned systemic therapy for those patients that were initiated with systemic therapy in stage BCLC stage C and improved to BCLC stage B under therapeutic systemic intervention, so they are possible candidates for TACE or local ablative treatments.

This shift of clinical stages under systemic therapy together with pathophysiological changes has made the approach of combining TACE with immunotherapy or TKIs increasingly intriguing. TACE induces acute hypoxia by embolizing tumor-feeding arteries, consequently triggering an elevated release of VEGF. Combining TKIs that target the VEGF signaling pathway with TACE presents an interesting strategy to counteract TACE-induced angiogenesis. Furthermore, a physiological rationale exists for combining with checkpoint inhibitors. TACE-induced tumor necrosis creates a hypoxic tumor environment, resulting in increased PD-L1 expression on both tumor and immune cells. Treatment with PD-1/PD-L1 inhibitors aims to counteract this effect accordingly [177].

The SPACE trial (phase III, NCT00855218) enrolled 307 patients who were randomly assigned to receive either TACE plus sorafenib or sorafenib alone. However, the study did not meet its primary endpoint, which was time to progression [178]. Similarly, the comparison between sorafenib alone and TACE plus sorafenib was addressed in the TACE-2 trial (phase III). In this study, 157 patients received TACE plus sorafenib, while 156 patients were treated with TACE only. There were no statistically significant differences between the two groups in terms of progression-free survival [179]. Contrarily, the TACTICS trial (phase II, NCT01217034) demonstrated a benefit for TACE plus sorafenib over TACE alone. Among 156 patients (*n* = 80 in the TACE plus sorafenib group, *n* = 76 in the TACE group), the mOS was 36.2 months with TACE plus sorafenib compared with 30.8 months with TACE alone, although the differences were not statistically significant. Additionally, the mPFS was 22.8 months with TACE plus sorafenib and 13.5 months with TACE monotherapy [180]. It is noteworthy that interpreting PFS in this study was complex due to the authors defining an individual PFS which diverged from the standard definition as it included ‘time to untreatable progression’. This evaluation encompassed not only radiological tumor response but also considerations of new extrahepatic spread, vascular invasion, and patient liver function to define ‘untreatable’ status.

The combination of TACE with lenvatinib was investigated in the TACTICS-L trial (phase II). This trial enrolled 62 patients, commencing lenvatinib treatment 2–3 weeks before TACE, pausing TKI treatment 2 days before and until 3 days after the intervention. Results showed an mPFS of 28.0 months (90% CI; 25.1–31.0) and an ORR of 88.7%, while the mOS was not reached [181].

Evidence supporting the combination of TACE with TKIs and immunotherapy arises from the retrospective CHANCE 001 trial, analyzing 826 patients with HCC who have received either TACE plus therapy with PD-(L)1 inhibitors and molecular target therapy (including TKIs) or TACE monotherapy. In the group undergoing TACE combined with systemic treatment, the mPFS was 9.5 months (95% CI; 8.4–11.0), notably improved compared to 8.0 months (95% CI; 6.6–9.5) in the TACE monotherapy group. TACE plus systemic treatment significantly improved mPFS (HR 0.70, *p* = 0.002) and displayed significant benefits in mOS and ORR. The combination group showed an mOS of 19.2 months (95% CI; 16.1–27.3) with an ORR of 60.1%. Conversely, the TACE monotherapy group showed an mOS of 25.7 months (95% CI; 13.0–20.2) and an ORR of 32.0% [182].

Additionally, the CISLC-12 trial (phase II, NCT05213221) reported promising safety and efficacy outcomes for the combined treatment of envafolimab, lenvatinib, and TACE, showing an ORR of 77.8% based on mRECIST criteria [183]. Similarly, TACE in combination with lenvatinib and sintilimab also demonstrated clinical efficacy in a retrospective study (mPFS: 13.3 months; mOS: 23.6 months) [184].

Another retrospective trial investigated the combination of lenvatinib plus camrelizumab plus TACE against lenvatinib plus TACE. The mPFS was 9.4 months for the triplet therapy and 5.9 months for the doublet therapy (*p* < 0.01). In addition, the authors reported that BLCLC stage is an independent prognostic factor in terms of overall survival and progression-free survival [185].

The combination or sequence of immunotherapy and TACE is interesting as a method for downstaging. In international guidelines, downstaging is currently defined as a pretreatment for HCC outside the Milan criteria with the aim of reducing the tumor size to defined selection criteria for liver transplantation. Established procedures include local ablation, surgical resection, or transarterial procedures (TACE, TARE). Due to the recent rapid development of systemic therapy for HCC and the evaluation of the anti-tumor efficacy of combinations of local ablative procedures or transarterial procedures with systemic therapy, downstaging through systemic therapy may also be possible in the future. Furthermore, the response to a downstaging therapy can be used as a tumor biological selection criterion. However, currently, there are only smaller studies or combination studies available that do not allow for evidence-based recommendation for downstaging through systemic therapy in international guidelines [14,186].

Recently, the data from the EMERALD-1 study (phase III, NCT03778957) were published. In total 616 pts with BCLC Stage A, B, and C were randomized to durvalumab plus TACE (Arm A: *n* = 207), durvalumab plus bevacizumab and TACE (Arm B: *n* = 204), or TACE alone (Arm C: *n* = 205). In a previous study, durvalumab plus bevacizumab showed promising clinical activity in patients with HCC (phase II, NCT02519348). The primary endpoint of EMERALD-1 was progression-free survival for treatment arm B vs. arm C. The triple combination of durvalumab plus bevacizumab and TACE demonstrated a significant prolonged mPFS of 15.0 months compared to TACE monotherapy, which showed an mPFS of 8.2 months (hazard ratio, 0.77; 95% CI; 0.61–0.98; *p* = 0.032) [81]. Treatment with durvalumab plus TACE (Arm A) was not superior to TACE alone [81]. The authors concluded that EMERALD-1 is the first phase III trial which showed improved anti-tumor efficacy with systemic therapy in combination with TACE in patients with HCC.

Furthermore, the ongoing DEMAND trial investigated the potential of atezolizumab plus bevacizumab in combination with TACE or local ablation in patients with intermediate-stage HCC. Patients were randomized in a 1:1 ratio and received either atezolizumab plus bevacizumab with sequential TACE upon progressive disease (Arm A) or atezolizumab plus bevacizumab simultaneously with TACE. The pending results of this study will also add value to assessing systemic therapy as an option for downstaging. It should be noted, however, that all patients included in the study had to be TACE-able from the beginning. Conversion from a status of non-TACE-able to TACE-able through the systemic therapy with atezolizumab/bevacizumab was therefore not possible [82].

Despite these encouraging results, conclusive evaluations of the combined TACE and TKI/CPI therapies will require further prospective randomized trials in the future (see Table 11).

### 4.5. Bispecific Antibodies (BsAbs) as Potential Therapeutic Agents for Systemic Therapy in HCC

A bispecific monoclonal antibody (BsAb) offers the unique ability to simultaneously bind to two different types of antigens or two distinct epitopes of the same antigen. One of the advantages of BsAbs is their capability to enhance the activity of immune cells while bringing them into direct contact with tumor cells by binding to tumor-specific antigens [187]. A subtype of BsAbs, known as bispecific T-cell engagers (BiTEs), functions by linking a CD3 antibody single-chain variable fragment with a tumor-associated antigen (TAA) or a tumor-specific single-chain variable fragment. BiTE antibodies can therefore be used to specifically control a T-cell-mediated immune response against certain target cells, such as tumor cells. For bispecific antibodies to be effective, the presence of highly specific antigens, which are preferably only present in the respective tumor cells, is necessary. The identification of such tumor-specific antigens in HCC is the subject of current research.

One target is the Epithelial Cell Adhesion Molecule (EpCAM), which plays a significant role in hepatocellular carcinoma (HCC). EpCAM overexpression has been linked to the promotion of tumor growth, progression and metastasis in HCC. EpCAMs are thought to participate in the sustenance of cancer stem cells (CSCs) within HCC. CSCs represent a minority subset of cancer cells endowed with self-renewal and tumor-initiating capabilities, driving tumor proliferation, recurrence, and therapy resistance. Notably, EpCAM-positive CSCs have been discerned in HCC, potentially exacerbating disease advancement and resistance to treatment [188,189,190]. Solitomab, a humanized bispecific EpCAM/CD3 antibody, demonstrated significant suppression of tumor proliferation in xenografts and the lysis of HCC cells in in vitro studies [191,192]. However, in a phase I study (NCT00635596) involving patients with advanced solid tumors, treatment with the EpCAM/CD3 BiTE solitomab was associated with dose-limiting toxicities. The study identified a maximum tolerated dose of 24 μg/day, which fell outside the therapeutic range. and 95% of patients showed side effects >grade 3, including diarrhea and elevated liver enzymes [193]. A crucial limitation of the effectiveness of EpCAM-targeted therapy is that only 15.9% to 48.7% of HCC cases are EpCAM positive [194]. Thus, predictive markers are necessary to identify patients who would benefit from such therapy.

Another target is Glypican 3 (GPC3), part of the glypican family, which adheres to the cell surface via a glycosylphosphatidylinositol anchor. Its overexpression is prevalent in HCC cases and is detected at heightened levels in the serum of many HCC patients. The upregulation of GPC3 expression significantly leads to tumor growth and forming metastases in HCC [195,196]. An immunohistochemical study revealed that GPC3 is expressed in 72% of HCC cases, while it is undetectable in hepatocytes from normal liver tissue [197]. While the exact mechanism of GPC3 signaling remains incompletely understood, its involvement in the canonical Wnt signaling pathway is suspected [198]. A BiTE targeting Glypican 3 (GPC3) and CD3 antigens (h8B-BsAb) demonstrated tumor regression in HCC xenograft mouse studies [199]. Furthermore, a bispecific NK-cell engager that targets CD16A of NK cells and GPC3 on HCC tumor cells has been shown to have anti-tumor activity in preclinical studies [200]. The BsAb GPC3/CD47 demonstrated anti-tumor efficacy in a HCC xenograft model [201]. Nonetheless, there are numerous imitations of GPC3-directed therapy. Only a subset of HCC patients expresses GPC3. This means that GPC3-directed therapy may not benefit all HCC patients. Even among GPC3-expressing HCC patients, there can be significant heterogeneity in GPC3 expression levels and tumor characteristics. This heterogeneity can impact the response to GPC3-directed therapy and contribute to treatment resistance.

Furthermore, a BiTE targeting transferrin receptor (TfR) and CD3 was investigated in a preclinical study and demonstrated anti-tumor efficacy in a HCC xenograft model [202]. The safety and anti-tumor efficacy of cadonilimab, an anti-PD-1/CTLA-4 bispecific antibody, was investigated in a phase 1b/2 basket trial for patients with advanced solid tumors (COMPASSION-03, NCT03852251). In total, 24 patients with hepatocellular carcinoma with at least one prior systemic treatment line were enrolled. Cadonilimab showed an mPFS of was 3.7 months (95% CI; 1.8–9.9) and an ORR of 16.7% (95% CI; 4.7–37.4), with a manageable safety profile [203].

Despite these promising preclinical findings, robust clinical data for the use of BsAbs in patients with HCC are currently limited. Further research through clinical trials is essential to explore and better understand the potential of BsAbs in the treatment of HCC. GPC3-directed therapy may need to be combined with other treatment modalities, such as chemotherapy or immunotherapy, to enhance patient outcomes (for ongoing studies, see Table 12).

### 4.6. Antibody–Drug Conjugates (ADCs) as Potential Therapeutic Agents for Systemic Therapy in HCC

Antibody–drug conjugates (ADCs) consist of two essential components. One function is a carrier molecule, typically a monoclonal antibody or its fragment, designed to selectively bind to cancer cells. This carrier molecule targets specific tumor antigens or receptors on the surface of cancer cells. The other function is the binding of a cytostatic drug.

Glypican-3 (GPC3) is a compelling target for antibody-based immunotherapies, being highly expressed in over 70% of HCCs but absent in normal adult tissue [197,204]. The internalization capability of GPC3 facilitates the utilization of ADCs for treating HCC. Preclinical studies on HCC cell lines suggest the cytotoxic effects of GPC3-targeted ADCs, such as hYP7-DC and hYP7-PC [205].

Another target for ADCs that has shown anti-tumor efficacy in preclinical studies is CD24 (G7mAb-DOX), a mucin-like molecule that is overabundant in a variety of human carcinomas [206,207]. Preclinical data for the efficacy of ADCs targeting CD147 (Anti-CD147 ILs-DOX), CD133 (AC133-vcMMAF) as well as c-Met (MetFab-DOX, anti-c-Met-IgG-OXA) and claudin 6 (CLDN6-DM1) also exist [208,209,210,211].

In the IMMU-132-01 basket trial (phase I/II, NCT01631552), sacituzumab govitecan, a Trop-2-directed ADC, was investigated in 495 patients with advanced solid tumors, including two patients with HCC. The reported ORR was 0% but the two patients with HCC reached a stable disease best response [212].

In terms of both anti-tumor activity and safety, there are still significant challenges to be overcome before the use of ADCs in the therapy of HCC becomes possible. A crucial challenge is the identification of the correct tumor-specific antigen (on taget toxicity). The chosen tumor-specific target antigen ought to exhibit a high cell surface expression. Following internalization into the HCC cell, it should undergo recycling back to the cell surface to sustain consistent expression, crucial for facilitating drug delivery to the cells [213]. Unselective antigens with high expression in healthy tissue could lead to increased off-target toxicity. Another challenge is the stability of the linkers. The linker must possess properties to rapidly and, above all, safely release the payload within the tumor cell (on target dilivery). Premature release (off-target delivery) leads to increased off-target toxicity and consequently limits therapy [214]. A major challenge and limitation in the treatment of HCC is the often present liver cirrhosis with impaired liver function and accompanying comorbidities (metabolic syndrome, alcohol dependency, etc.). ADCs are metabolized via hepatic and renal pathways. Reduced liver function in the presence of liver cirrhosis thus leads consecutively to increased therapy-associated toxicity. Safety data for the use of ADCs in patients with HCC are not available. However, information on hepatotoxicity is available from other tumor entities. Hepatotoxicity depends crucially on the selected payload class. A meta-analysis (with 43 studies) showed relevant hepatotoxicity primarily for payload class DM1, with relevant hepatotoxicity (grade 3/4) occurring in up to 20% of patients [215]. Furthermore, the development of resistance to both the cytotoxic payload and the carrier molecule, for example due to secondary mutations of the tumor-specific target structure, represents a therapeutic limitation [216].

Given the highly vascularized nature of HCC tumors, there are opportunities for treatment strategies combining TACE and local arterial administration of ADCs.

However, further clinical trials are needed to evaluate the efficacy and safety of ADCs in patients with advanced HCC, particularly regarding liver function. Ongoing studies are reported in Table 13.

### 4.7. Targeting Cyclin-Dependent Kinase 4/6 (CDK4/6) for Systemic Treatment of HCC

Cyclin-dependent kinases CDK4 and CDK6 play crucial roles in cell cycle control. The serine/threonine protein kinases CDK4 and CDK 6 are pivotal in regulating the transition from the G1 phase to the S phase by inhibiting the retinoblastoma protein (pRb).

Palbociclib, a selective inhibitor of CDK4/6, has demonstrated anti-tumor efficacy in preclinical models involving human liver cancer cell lines [217]. Additionally, a combination of palbociclib and regorafenib has shown anti-tumor activity in hepatocarcinoma cell lines [218]. Preclinical studies in hepatocellular cell lines have also shown increased anti-tumor activity for lenvatinib in combination with abemaciclib compared to the use of the respective single substances alone [219].

Clinical evidence exists from a phase II trial (NCT01356628) where patients received palbociclib after failure of first-line therapy. In total, 21 patents were enrolled and treated with a daily dose of 125 mg palbociclib. The mOS was 10.5 weeks, and median time to progression (mTTP) was 24 weeks [220]. Currently, the use of palbociclib is under further investigation in a phase II basket trial (MATCH trial, NCT02465060). Another CDK4/6 inhibitor, milciclib, underwent evaluation in a phase II trial (NCT03109886) and demonstrated anti-tumor efficacy in patients with unresectable or metastatic HCC. The primary endpoint, clinical benefit rate (CBR), was met with a CBR of 68% [221]. Ribociclib in combination with TACE was investigated in a phase II trial (NCT02524119), although results from this study have not been reported yet.

In conclusion, further studies are necessary to determine the precise role of CDK4/6 inhibitors in the therapy of HCC. The CDK4/6 and PI3K/AKT/mTOR signaling pathways have several interactions. Activation of the PI3K/AKT/mTOR pathway leads to increased levels of cyclin D1, which binds to CDK4 and CDK6 and thereby activates them [222]. From the results of preclinical research, a combination of CDK4/6 inhibitors with TKIs appears to be promising and should be pursued in further clinical studies. CDK4/6 inhibitors can modulate the tumor microenvironment and enhance anti-tumor immune responses. Combining CDK4/6 inhibitors with immunotherapy agents, such as immune checkpoint inhibitors, may potentiate immune-mediated tumor regression in HCC [223,224].

Primary and secondary resistance mechanisms represent a challenge in the use of CD4/6 inhibitors. The retinoblastoma gene (RB1) is mutated in 3–8% of patients with HCC [225]. It has been shown that the loss of function of retinoblastom gene (RB1) correlates with resistance to palbociclib in human liver cancer cell lines [217].

The MATCH Screening Trial (phase II, NCT02465060) actually examines the effects of palbociclib in patients with HCC and other cancers harboring CDK4 or CDK6 amplification and preserved function Rb protein.

### 4.8. Targeting Epidermal Growth Factor Receptor (EGFR) for Systemic Treatment of HCC

EGFR is a membrane-bound receptor tyrosine kinase. EGFR activation initiates downstream signaling cascades that regulate cell cycle progression, cell survival, and angiogenesis. In HCC, overexpression or dysregulation of EGFR can prompt uncontrolled cell growth and division, thereby contributing to tumor development and progression. Upon binding of ligands such as transforming growth factor α (TGF-α) and epidermal growth factor (EGF), signaling pathways, including RAS/RAF/MEK/ERK and PI3K/AKT/Mtor, are activated. Consequently, EGFR overexpression promotes cell proliferation, inhibits apoptosis, and supports tumor angiogenesis, which collectively contribute to the progression of HCC [226]. EGFR overexpression occurs in up to 68% of human HCC cases and strongly correlates with poor prognosis [227].

Erlotinib functions as a selective inhibitor of the tyrosine kinase domain of the EGFR. A phase II trial (NCT00881751) investigated the efficacy of erlotinib combined with bevacizumab versus sorafenib in treatment-naive patients with advanced HCC. In total, 90 patients were enrolled and randomly assigned to receive erlotinib (150 mg once daily) plus bevacizumab (10 mg/kg every 2 weeks) or sorafenib. No significant difference was observed in mOS between both groups. The mOS was 8.55 months (95% CI; 7.00–13.9) for patients treated in the combination group and 8.55 months (95% CI; 5.69–12.2) for patients in the control arm [228]. Another single-arm phase II trial (NCT00365391) evaluating erlotinib plus bevacizumab in patients with advanced HCC (first- or second-line setting) failed to meet the primary endpoint [229]. Investigating erlotinib plus bevacizumab after sorafenib therapy also showed no significant activity in unselected patients previously treated with sorafenib [230]. However, a single-arm phase II trial (NCT01180959) showed promising signals for clinical benefit with erlotinib plus bevacizumab in second-line therapy. In total, 44 patients in second-line setting were enrolled and demonstrated an mOS of 9.9 months (95% CI; 8.3–15.5) [231].

The SEARCH trial (phase III) aimed to assess the benefits of adding erlotinib therapy to standard sorafenib therapy in patients with treatment-naive HCC. In total, 720 patients were enrolled and randomly assigned to receive erlotinib plus sorafenib or sorafenib plus placebo. The mOS was similar in both groups, with a trend toward higher ORR observed in the group treated with erlotinib plus sorafenib [232].

Cetuximab, a monoclonal mAb against EGFR, failed to demonstrate anti-tumor efficacy as a single agent in patients with advanced HCC and a maximum of 2 prior systemic treatment regimens (phase II trial, NCT00142428). None of the 30 patients achieved a tumor response (ORR 0%), and mOS was 9.6 months (95% CI; 4.3–12.1 months) with an mPFS of 1.4 months (95% CI; 1.2–2.6 months) [233]. Cetuximab was also studied in combination with CAPOX and GEMOX, but no clinically meaningful benefit in terms of tumor response and overall survival was demonstrated compared to the established survival data for sorafenib [234,235].

The use of EGFR inhibitors as a single substance did not prove to be promising, partly due to the development of resistance mechanisms. Resistance mechanisms include EGFR mutations which lead to constitutive activation of the EGFR pathway or activation of alternative signaling pathways that bypass EGFR dependence. Further research on combination therapies could therefore help to increase anti-tumor activity. It is known that HCC cells develop resistance to lenvatinib by activating EGFR via several downstream signaling pathways [236,237]. The inhibition of FGFR by lenvatinib treatment leads to feedback activation of the EGFR-PAK2-ERK5 signaling pathway, which can be blocked by EGFR inhibition. Combining an EGFR inhibitor (gefitinib) with lenvatinib demonstrated powerful anti-cancer effects both in vitro and in various mouse models of liver cancer. In a clinical trial (phase I, NCT04642547), 12 patients with advanced HCC who were unresponsive to lenvatinib were enrolled and treated with gefitinib plus lenvatinib. The authors reported meaningful clinical responses [237].

Another therapeutic approach to address the EGFR signaling cascade is the blockade of downstream molecular structures such as PI3K (copanlisib), AKT (MK-2206), or mTOR (sirolimus, everolimu, AZD8055, onatasertib). Both monotherapy and combination therapy with EGFR inhibitors are conceivable here. Approaches discussing the PI3K/AKT/mTOR axis are discussed in Section 4.10. Furthermore, the EGFR signaling pathway plays a crucial role in regulating the tumor microenvironment and the recruitment of inflammatory cells [226]. Therefore, combination strategies with immunocheckpoint inhibition also appear possible.

Presently, the available data do not support recommending the use of erlotinib or cetuximab in HCC therapy. Further clinical trials focusing on combination strategies are necessary. The MATCH Screening Trial (phase II, NCT02465060) is a clinical trial which will examine the effects of afatinib and osimertinib in patients with HCC and other cancers harboring EGFR-activating mutations.

### 4.9. Targeting ROS1/ALK/MET Alterations for Systemic Treatment of HCC

The ROS1, ALK, and MET genes encode for receptor tyrosine kinases, playing essential roles in various cellular processes. c-Met is a receptor tyrosine kinase encoded by the MET gene that binds hepatocyte growth factor (HGF). Activation of this receptor induces multiple downstream signaling pathways (such as RAS/RAF/MEK/ERK and PI3K/AKT/mTOR) regulating cell proliferation, cytoskeleton reorganization, cell invasion and cell migration. Overexpression of c-Met has been identified as an independent risk factor with poor prognosis in patients with HCC. Gene amplification or mutations in MET can also lead to enhanced activation of c-Met and its downstream signaling pathways. Therefore, targeting and inhibition of c-Met could serve as a viable molecular target, especially for selected patients [238].

ROS1 gene rearrangements are relatively rare in HCC compared to other cancers like lung cancer. ALK gene alterations in HCC include copy number variations. However, when present, especially ROS1 alterations can contribute to the activation of signaling pathways (such a JAK/STAT pathway) that promote cell growth and division [239,240].

The c-Met receptor tyrosine kinase inhibitor tivotinib was investigated in a phase II trial (NCT00988741) involving patients with advanced HCC who had previously failed at least one prior systemic treatment. In this study, 71 patients were randomly assigned in a 2:1 ratio to receive tivantinib (360 mg twice daily, later reduced to 240 mg because of toxicity) or placebo. The trial indicated a slight but statistically significant difference in median time to progression (TTP) between the tivantinib and the placebo group: 1.6 months (95% CI; 1.4–2.8) for tivantinib and 1.4 months (95% CI; 1.4–1.5) for placebo. In a subgroup analysis, patients with MET high-expression tumors treated with tivantinib showed a median TTP of 2.7 months (95% CI; 1.4–8.5) compared to 1.4 months (95% CI; 1.4–1.6) for those receiving placebo [241]. However, in a randomized phase III trial (METIV-HCC, NCT01755767) involving 340 patients with HCC and high MET expression who had failed prior treatment with sorafenib, tivantinib did not demonstrate improved OS. Patients were randomly assigned in a 2:1 ratio to receive tivantinib or placebo. The reported mOS was 8.4 months (95% CI; 6.8–10.0) in the tivantinib group and 9.1 months (95% CI; 7.3–10.4) in the placebo group [242]. Similarly, another phase III trial (JET-HCC, NCT02029157) also showed no benefit for c-Met inhibition in MET-high advanced HCC [243].

Capmatinib, a selective c-Met inhibitor, showed an ORR of 30% in a single-arm phase II study (NCT01737827) that enrolled patients with advanced HCC and high MET expression (1 complete response, 2 partial responses out of 10 patients) [244]. However, combining capmatinib with spartalizumab (PD-1 inhibitor) versus spartalizumab monotherapy failed to enhance the ORR in a phase I/II trial (NCT02795429).

A phase Ib/II study (NCT02082210) investigating the combination of ramucirumab plus emibetuzumab (anti-MET mAb) in 45 patients with HCC demonstrated an ORR of 6.7% and a PFS of 5.42 months (95% CI; 1.64–8.12). Notably, HCC with high MET expression showed improved PFS compared to HCC with low MET expression (mPFS: 8.1 months versus 2.8 months) [245].

The c-Met inhibitor tepotinib demonstrated improved time to progression compared to sorafenib in treatment-naive patients with advanced HCC [246]. Additionally, the anti-ALK-1 mAb GT90001 exhibited anti-tumor efficacy in combination with nivolumab for treating patients with advanced HCC after failure of first line therapy (NCT03893695) [247].

There is evidence for the efficacy of c-Met inhibitors in HCC. However, their use should currently only be discussed after established treatment options have been exhausted and MET overexpression has been evaluated. Future direction for c-MET inhibitors in HCC therapy involves refining strategies, identifying suitable patient populations, and exploring combinations with other agents. Crizotinib for patients with ROS1/ALK/MET alterations is actually under investigation in the NCI-MATCH trial (see Table 14).

### 4.10. Targeting the PI3K/Akt/mTOR Signaling Pathway for Systemic Treatment of HCC

The PI3K/Akt/mTOR signaling pathway leads to tumor development in HCC via several pathophysiological mechanisms. Active PI3K converts phosphatidylinositol-4,5-bisphosphate (PIP2) to phosphatidylinositol-3,4,5-triphosphate (PIP3). PIP3 then recruits proteins like 3-phosphoinositide-dependent protein kinase 1 (PDK1) and Akt to the cell membrane. Akt, also known as protein kinase B (PKB), is a key enzyme involved in multiple cellular processes. Fully activated Akt stimulates downstream proteins, including mammalian target of rapamycin (mTOR) [248]. Activation of the PI3K/AKT/mTOR pathway promotes cell growth and division, a hallmark of cancer. Dysregulation in this pathway can lead to uncontrolled cell proliferation, contributing to the formation and growth of HCC tumors. Additionally, the pathway regulates cell survival by inhibiting apoptosis (programmed cell death). When excessively activated, it prevents cells from undergoing programmed cell death, allowing cancer cells to survive and multiply uncontrollably. The pathway also influences angiogenesis, which is crucial for tumor growth and metastasis. It stimulates the production of several factors promoting blood vessel formation, facilitating the supply of nutrients and oxygen to the tumor [248].

Activation of this pathway can occur through various growth factors and receptors, such as the epidermal growth factor receptor (EGFR) and insulin-like growth factor receptor (IGF-1R), which are often overexpressed or dysregulated in HCC. Genetic mutations or aberrant activation of components within this pathway (e.g., mutations in PI3KCA, loss of PTEN) can lead to persistent activation, contributing to HCC development and progression. Dysregulated PI3K/Akt/mTOR signaling is associated with resistance to certain cancer therapies, making it a challenging target in the treatment of HCC [248].

The PI3K/Akt/mTOR pathway exhibits aberrant activation in approximately 50% of patients with HCC [248]. Since the activation of the PI3K/AKT/mTOR signaling pathway primarily occurs via EGFR, therapeutic approaches to block EGFR have been pursued. The efficacy and safety of combining bevacizumab and erlotinib, an EGFR inhibitor, were investigated in an open-label phase II trial (NCT01180959). The authors reported that erlotinib can inhibit the activation of the Akt signaling pathway, indicating comparable efficacy between the combination therapy and sorafenib monotherapy [231].

It is known that currently available multi-TKIs and VEGF inhibitors activate not only the RAS/RAF/MEK/ERK signaling pathway but also the PI3K/Akt/mTOR pathway. Therefore, combinations of TKIs with mTOR inhibitors and targets downstream in the RAS/RAF/MEK/ERK signaling pathway have recently been investigated [248]. Rapamycin (sirolimus) in combination with bevacizumab (phase I, NCT00467194) in patients with unresectable HCC showed evidence of anti-vascular activity, along with promising clinical activity [249]. The combination of bevacizumab and everolimus was also being investigated in a phase II trial (NCT00775073), but currently, no results have been reported.

Sirolimus, an mTOR inhibitor, underwent investigation in phase I and II trials involving patients with advanced HCC. In a phase I trial with 21 patients and advanced HCC, sirolimus was studied aiming for serum levels between 4 and 15 µg/mL. Among the participants, one patient achieved a partial response, and five achieved stable disease. The mOS was 6.5 months [250]. In a single-arm phase II study, an ORR of 8% (95% CI; 0.98–26.03) was reported, with an mOS of 26.4 weeks and an mPFS of 15.3 weeks [251]. However, despite the formal positivity of the study, the observed results fell short clinically compared to those achieved with currently established therapies. Nevertheless, the combination strategy of sirolimus added to a VEGF inhibitor (phase I, NCT00467194) appears to have anti-tumor efficacy [249].

The mTOR inhibitor everolimus did not demonstrate significant clinical benefit in the treatment of HCC. In the EVOLVE-1 trial (phase III, NCT01035229), 546 patients diagnosed with HCC at BCLC stage B or C, following sorafenib treatment failure, were randomly assigned in a 2:1 ratio to receive everolimus or a placebo. The mOS was 7.6 months with everolimus and 7.3 months with placebo, without demonstrating a statistically significant difference [252]. Additionally, in a randomized phase II trial (NCT01005199), the combination of sorafenib with everolimus was compared with sorafenib monotherapy. The findings did not indicate evidence supporting the superiority of sorafenib plus everolimus over sorafenib alone [253].

A phase I and phase I/II study (NCT01008917, NCT01687673) of temsirolimus in combination with sorafenib investigated the anti-tumor efficacy in patients witch advanced HCC and demonstrated an mOS of 8.8 months (95% CI; 6.8–14.8). However, the expectations and results fell short of the historical standard of the SHARP study.

Understanding the significance of the PI3K/AKT/mTOR pathway in HCC pathogenesis has led to the development of targeted therapies aiming the inhibition of this pathway. Various inhibitors targeting components of this pathway are being investigated in clinical trials as potential treatments for HCC (see Table 15). However, due to the complexity and cross-talk with other signaling pathways, the efficacy of these therapies in HCC treatment remains an area of ongoing research and development. Because the PI3K/Akt/mTOR pathway plays a critical role in multiple cellular processes, efforts to target it can result in adverse events significant enough to prompt the cancellation of treatment.

Furthermore, resistance to such treatments has been observed. In our view, addressing individual target structures of the PI3K/Akt/mTOR signaling pathway is not expedient in the treatment of HCC. Rather, future research and studies must focus on addressing target structures at several levels of the PI3K/Akt/mTOR signaling pathway and on the simultaneous addressing of alternating signaling pathways such as RAS/RAF/MEK/ERK [254]. Also, understanding the feedback mechanisms in the PI3K/Akt/mTOR pathway is crucial for developing effective therapeutic strategies in treatment of HCC.

The novel PI3K inhibitor copanlisib demonstrated anti-tumor activity in an HCC cell line [255]. In addition, the Akt inhibitor capivasertib also showed anti-tumor efficacy in preclinical studies [256]. Both substances are currently being investigated in a phase II clinical trial (see Table 15).

Further possibilities to increase the effectiveness of therapy with mTOR inhibitors by applying the substances effectively into the tumor tissue are approaches with nanoparticle-bound substances [257].

### 4.11. Targeting the RAS/RAF/MEK/ERK Signaling Pathway for Systemic Treatment of HCC

Evidence suggests that the RAS/RAF/MEK/ERK pathway is activated in up to 50% of patients with HCC [258]. The signaling pathway plays a pivotal role in the regulating various cellular mechanisms, including cell proliferation, cell differentiation, the cell cycle, and apoptosis. Dysregulation of the RAS/RAF/MEK/ERK pathway has been linked to oncogenesis in HCC, evident through its stimulation of cell growth, cell survival, cell motility, and cell proliferation. [259] Furthermore, activation of RAS/RAF/MEK/ERK leads to extracellular matrix degradation and enhances tumor cell invasion and metastasis [259].

Selumetinib, an oral competitive MEK1/2 inhibitor, underwent investigation in a phase II trial (NCT00604721) including patients with locally advanced HCC. In total, 19 patients were enrolled and received 100 mg of selumetinib twice daily. The trial was halted at the first interim analysis as no patients demonstrated a radiographic tumor response. The median time to progression was 8 weeks, the mPFS was 1.4 months (95% CI; 1.2–2.5 months) and the mOS was 4.2 months (95% CI; 1.9–6.0) [260]. The combination of sorafenib and selumetinib was investigated in a single-arm phase I trial (NCT01029418). Among 27 patients, 4 demonstrated a partial response (15%) and 13 patients (48%) showed a stable disease. The mOS was 14.4 months and mPFS was 5.6 months [261]. This combination is intriguing for synergistic effects as sorafenib treatment can increase ERK activity, and MEK inhibition with selumetinib aims to counteract this process [262]. Sorafenib, acting as an inhibitor of RAF kinases, can induce upregulation of the downstream RAS/RAF/MEK/ERK pathway in cancer cells [263]. Nevertheless, trametinib (another MEK inhibitor) in combination with sorafenib demonstrated only limited anti-tumor efficacy in a phase I trial [264].

Another MEK1/2 inhibitor, refametinib, was evaluated in a phase II study (NCT01915602) involving patients with advanced HCC carrying RAS mutations. In total, 70 patients were enrolled and received refametinib 50 mg in addition to twice-daily sorafenib (morning dose: 200 mg; evening dose: 400 mg). The mOS was 290 days (approximately 10 months), with the best responders showing RAS mutations [265]. In a retrospective analysis of two prospective phase II studies (refametinib monotherapy; refametinib monotherapy versus sorafenib plus refametinib), 1.318 patients with advanced HCC were screened for RAS mutations. In the collective 59 patients, 4.4% carried a RAS mutation. Among them, 16 received refametinib, and 16 patients received refametinib plus sorafenib. The ORR in the combination cohort was 6.3%, compared with 0% in the monotherapy cohort. In the combination cohort, the mOS was 12.7 months and the mPFS was 1.5 months, while in the monotherapy cohort, the mOS was 5.8 months and the mPFS was 1.9 months [266].

A problem with treating tumors using individual targeted substances is the development of resistance mechanisms, such as the formation of secondary mutations or compensatory activation of alternative signaling pathways. For the treatment of HCC with MEK inhibitors, combination approaches with TKIs (inhibition of structures upstream) or PI3K inhibitors or mTOR inhibitors (inhibition of crosstalk to the PI3K/Akt/mTOR signaling pathway) appear sensible. Survival data of phase I/II evidence indicate potential efficacy for the TKI and MEK1/2 inhibitor combination, yet additional phase III studies are essential to evaluate their benefit in treating patients with advanced HCC [261,265]. The combination of MEK inhibitors and mTOR inhibitors showed evidence of efficacy in preclinical in vitro and in vivo models [261,267]. Furthermore, combining MEK inhibitors with immunotherapy, such as immune checkpoint inhibitors, may enhance the anti-tumor immune response due to the property of MEK inhibitors which may modulate the tumor microenvironment, making it more conducive to immunotherapy.

### 4.12. Targeting the Wnt/β-Catenin Signaling Pathway for Systemic Treatment of HCC

The Wnt/β-catenin pathway is crucial in regulating various cellular processes like cell proliferation, differentiation, and embryonic development. Dysregulation or aberrant activation of this pathway is frequently observed in HCC, contributing to tumor initiation, progression, and metastasis. Several strategies are being explored to target this pathway, such as inhibition of WnT ligands or targeting β-catenin [268].

There are some preclinical data on the efficacy of substances targeting the Wnt/β-catenin pathway in HCC cell lines. These approaches include substances inhibiting the interaction of β-catenin with the T-cell factor (TCF), such as the fungal derivatives PKF115-854, PKF118-310, and CGP049090 [269]. These substances have shown inhibitory effects on the growth of HCC cells. The novel inhibitor of Wnt ligands CGX1321 was investigated in a phase I/Ib trial in patients with advanced solid gastrointestinal tumors (NCT02675946, NCT03507998). The study enrolled 77 patients, including 38 patients with solid tumors in the phase I CGX1321 dose escalation part. The authors reported that CGX1321 has shown powerful inhibition of the WNT pathway with tolerable side effects. Nevertheless, it must be emphasized that in the phase Ib part, mainly patients with colorectal carcinoma or small-bowel carcinoma were included. Additionally, it is unclear how many patients with HCC were included in the dose escalation part and dose expansion part of the phase I study. Subgroup analyses are pending [270].

Targeting the Wnt/β-catenin pathway in HCC faces challenges, including the complexity of the pathway, its crucial role in normal physiological processes, and potential toxicities associated with systemic inhibition [268].

However, further preclinical studies and clinical trials are essential to validate the efficacy, safety, and clinical utility of Wnt/β-catenin-targeted therapies in HCC treatment.

### 4.13. Targeting Dickkopf-1 (DKK-1) for Systemic Treatment of HCC

Targeting Dickkopf-1 (DKK-1) in hepatocellular carcinoma (HCC) has emerged as a potential therapeutic strategy due to its involvement in the regulation of the Wnt/β-catenin signaling pathway. DKK-1 is a secreted protein that modulates the Wnt/β-catenin pathway by binding to LRP5/6 co-receptors, preventing the formation of the Wnt-Frizzled-LRP complex and inhibiting downstream β-catenin activation [271,272]. Recent studies have reported that DKK-1 is linked to carcinogenesis and poor prognosis in HCC [273].

Furthermore, there is evidence that DKK-1 influences the risk of metastases and the risk of tumor recurrence [274,275]. The specific molecular mechanisms by which DKK-1 contributes to tumorigenesis have not yet been conclusively clarified. Since DKK-1 is physiologically an inhibitor of the Wnt/β-catenin signaling pathway, alternative signaling cascades or crosstalk between different signaling cascades may play a decisive role.

The monoclonal antibody DKN-01 is a potential therapeutic agent targeting DKK1 and is under extensive investigation [276]. DKN-01 neutralizes free DKK1 from the tumor-microenvironment. Interestingly, preclinical studies in HCC cell lines have shown that inhibition of DKK-1 may enhance the anti-tumor activity of sorafenib [277]. Further research and clinical trials are necessary to explore safety and efficacy of drugs targeting DKK-1 as a potential treatment option in HCC (see Table 16).

There is also evidence for the use of DKK-1 as a diagnostic biomarker. In a previous study, the determination of DKK-1 in serum was shown to have a high sensitivity and specificity with regard to the detection of HCC [278].

### 4.14. Targeting Neurotrophe Tyrosine Receptor Kinase (NTRK) for Systemic Treatment of HCC

The tropomyosin receptor kinase (TRK) family consists of three members—TRK A, TRK B, and TRK C—predominantly expressed in human nervous tissue [279]. These receptors are encoded by the NTRK 1, NTRK 2, and NTRK 3 genes. The primary cause of oncogenic TRK activation involves NTRK gene fusions, resulting in constant activation of the TRK kinase domain independent of ligands [280]. This sustained activation triggers signaling pathways like MAP kinase, protein kinase C (PKC), and phosphatidylinositol 3-kinase (PI3K), ultimately promoting cell proliferation or hindering apoptosis signaling [281].

Larotrectinib is approved by the FDA and EMA for treatment of patients with solid tumors harboring NTRK gene fusion. Notably, the basket trials leading to the approval of larotrectinib included only one patient with hepatocellular carcinoma and NTRK gene fusion. Larotrectinib for patients with NTRK alterations is actually under investigation in the NCI-MATCH Trial (see Table 17).

Entrectinib is another potent inhibitor of TRK A/B/C and is approved by both the FDA and EMA for use in adult and pediatric patients with solid tumors carrying an NTRK gene fusion based on the results of the pooled analysis of three phase 1 or 2 clinical trials (ALKA-372–001 NCT03066661, STARTRK-1 NCT02097810, and STARTRK-2 NCT02568267).

### 4.15. Adoptive Cell Transfer (ACT) as Potential Treatment Option in HCC

Adoptive cell transfer (ACT) involves the autologous or allogeneic transfer of living cells, often immune cells, particularly for anti-tumor therapies. This technique enables the genetic modification of immune cells ex vivo. Consequently, the immune system adopts these transgenic cells, such as T cells designed to recognize specific tumor antigens, acquiring new immunological properties. ACT is notably more specific than a classic systemic chemotherapy, targeted therapy, or CPI. NK cells or dendritic cells can be utilized for adoptive cell transfer as well. In contrast to antibodies or other targeted drugs, ACT can be activated and replicated in vivo, leading to a prolonged and durable anti-tumor effect [282,283].

The use of tumor-infiltrating lymphocytes (TILs) in treating HCC has been investigated across several clinical trials. The approach involves isolating TILs during tumor resection, stimulating and replicating them ex vivo in the presence of anti-CD3 antibody and interleukin-2 (IL-2), and subsequently reintroducing them to the patient following lymphocyte depletion [284]. In a randomized trial which enrolled 150 patients with early-stage HCC treated by surgical resection, patients were randomly assigned to receive either TIL-ACT (injection of lymphocyte infusions) or no adjuvant therapy (controls). While the TIL-ACT group showed significantly prolonged recurrence-free survival (38% versus 22% at 5 years), no significant differences were observed in overall survival [285]. A phase I clinical trial (NCT01462903) demonstrated manageable toxicity profiles in patients with HCC treated with TILs post-resection [286]. Additionally, studies revealed that increased tumor infiltration by lymphocytes correlated with a better prognosis in terms of recurrence rates in HCC [287]. Another clinical trial also indicated that higher concentrations of CD8+ lymphocytes were associated with improved overall survival post-HCC resection [288].

However, using TILs in HCC therapy remains challenging. There is an upregulation of immune checkpoints post-TIL isolation, potentially leading to decreased effectiveness due to reduced immunostimulatory effects. Another challenge involves obtaining a sufficient quantity of TILs from tumor tissue. Given these technical challenges, using lymphocytes from peripheral blood has emerged as an alternative T-cell source.

Cytokine-induced killer cells (CIKs) are immune cells generated ex vivo by expanding peripheral blood mononuclear cells (PBMCs) in the presence of IL-2, INF-alpha, and anti-CD3 monoclonal antibodies. CIKs comprise NKT cells, cytotoxic T cells, and NK cells [289]. Their advantage includes simplified sample collection, rapid proliferation, and high therapeutic efficacy against cancer cells with minimal toxicity towards normal cells.

A phase I trial assessing the safety of CIK therapy for HCC treatment demonstrated no significant side effects [290]. In a randomized phase III trial (NCT00699816), adjuvant CIK therapy after tumor resection for HCC demonstrated benefit in reducing the recurrence rate. The study, involving 230 patients with HCC treated by surgical resection, radiofrequency ablation, or percutaneous ethanol injection, randomly assigned patients to receive autologous CIK cell injection or no adjuvant therapy. The median time of recurrence-free survival (RFS) was 44.0 months for the immunotherapy group compared with 30.0 months for the control group [291]. A 5-year follow-up confirmed the sustained efficacy of adjuvant CIK immunotherapy, showing significant improvement in both RFS and OS [292]. Another phase III trial (NCT00769106) treating patients with HCC post-resection with CIKs versus no adjuvant treatment demonstrated a benefit in median time to tumor recurrence (TTR) but not in OS [293]. All patients in the CIK group completed treatment as per the protocol. The CIK group demonstrated a median time to recurrence (TTR) of 13.6 months compared to 7.8 months in the control group (*p* = 0.01), while no significant differences were observed in DFS and OS between the groups. A randomized clinical trial compared the efficacy of combining CIKs with TACE versus TACE alone, revealing an advantage in terms of OS with the CIK combination [294]. Additionally, a randomized phase II trial evaluated CIKs in combination with standard treatment versus standard treatment alone. This trial enrolled 132 treatment-naive patients randomly assigned to either arm A (CIKs plus standard therapy) or arm B (standard therapy alone). Standard therapy included TACE, resection, or best supportive care (BSC). The reported results demonstrated significantly higher mOS and mPFS in patients from arm A compared to those in arm B [295].

Two retrospective studies evaluated the efficacy of autologous CIK cell transfusion in combination with TACE and/or RFA compared to therapy with TACE and/or RFA in patients with HCC. Patients receiving CIK treatment demonstrated significantly longer mOS than those in the control group [296,297].

Immuncell-LC, the only commercially available CIK agent, serves as adjuvant therapy for patients after curative treatment. It received approval from the Korea Food and Drug Administration (KFDA) in 2007 and obtained orphan drug designation from the FDA for the treatment of HCC in 2018.

In summary, there is clinical evidence supporting the efficacy of CIKs in HCC treatment. However, further studies are necessary to determine the status of CIKs compared to established treatment options and to identify the specific patient population that would benefit most from CIK therapy (see Table 18).

Cancer therapy using chimeric antigen receptor T (CAR-T) cells represents a novel approach in immunotherapy for different entities. Chimeric antigen receptors are comprised of an extracellular antigen-binding domain, typically an antibody-derived single-chain variable fragment that combines the variable heavy and light chains from a monoclonal antibody specific to the tumor-associated antigen. They also feature a transmembrane domain and an intracellular domain composed of activating components from the T-cell receptor complex, serving as the signaling domain [298]. In CAR-T-cell therapy, T cells are genetically engineered ex vivo to recognize tumor-associated antigens, leading to targeted destruction of tumor cells. However, for CAR-T-cell therapy targeting malignant solid tumors, the primary challenge lies in identifying tumor-specific antigens, an area of ongoing research.

Preclinical studies exploring CAR-T-cell therapy targeting various HCC-associated antigens have demonstrated promising results [299]. One of the extensively studied targets in CAR-T-cell therapy is GPC3, primarily expressed in HCC cells and minimally or not expressed in normal tissue [300]. Studies on GPC3 CAR-T cells have demonstrated their anti-tumor effectiveness through both in vitro and in vivo experiments, including combination therapies involving sorafenib [301,302]. Early clinical phase I trials (NCT02395250, NCT03146234) provided evidence supporting the anti-tumor efficacy and safety of GPC3 CAR-T cells. Among 13 patients treated with GPC3 CAR-T therapy, 2 showed partial responses, and 1 exhibited stable disease. The reported median overall survival (mOS) was 278 days (95% CI; 48–615 days) [290]. Additionally, these trials highlighted better patient outcomes in cases when lymphodepletion was performed before CAR-T-cell infusion. In another phase I trial (NCT03980288) focusing on fourth-generation GPC3 CAR-T cells, six patients with HBV-related metastatic HCC participated, demonstrating a reported ORR of 16.7% and an mPFS of 4.2 months [303]. CAR-T cells expressing IL-7 and CCL19 to improve efficacy of proliferation and migration were investigated in a phase I clinical trial (NCT03198546). One patient with HCC (GPC3 positive) demonstrated a complete elimination of the tumor within 30 days after intra-tumor injection [304].

AFP represents another prominent target expressed in numerous cases of HCC. However, CAR-T cells face a physiological challenge in recognizing intracellularly located antigens like AFP. Specialized CAR-T cells have been engineered to address this issue by targeting the AFP-MHC complex, which presents intracellular antigens through MHC class 1 molecules [305]. These modified AFP-MHC CAR-T cells have exhibited promising anti-tumor activity in preclinical studies both in vitro and in vivo [306]. A phase I trial (NCT03349255) evaluating the safety and efficacy of AFP-MHC CAR-T cells demonstrated a manageable safety profile and reported tumor size reduction in three out of six patients. However, the trial was terminated prematurely due to the evaluation of a new T-cell construct for the same indication.

Additionally, positive preclinical in vitro and in vivo data exist for CAR-T cells targeting c-Met, CD147, CD133, and the natural killer group 2 member D ligand (NKG2DL) [307,308,309]. During a clinical phase II trial (NCT02541370), the anti-tumor efficacy of CAR-T-133 cells was investigated in patients with advanced HCC, and 14 out of 21 patients experienced disease stability with an mPFS of 6.8 months and an mOS of 12 months [310].

Adverse effects of CAR-T-cell therapy in solid tumors like HCC arise from challenges such as the lack of the highly specific tumor antigen evaluation, on the one hand, and tumor antigen heterogeneity on the other hand. Furthermore, there are difficulties related to limited transfer and infiltration of infused CAR-T cells into tumor tissues, which may lead to cross-reactivity causing on-target, off-tumor toxicity. The immunosuppressive nature of the tumor microenvironment in HCC poses another challenge, potentially resulting in the rapid loss of CAR-T-cell effectiveness. Despite these obstacles, several phase I and II studies are underway, investigating the efficacy of CAR-T-cell therapy in patients with HCC (see Table 19).

A challenge of CAR-T cell therapy is increasing efficacy while ensuring a safe and well-tolerated application. One approach to this is the design of novel CAR-T cells, which, through the activation of co-stimulatory pathways and increased cytokine secretion, induce an enhanced T-cell response. The activation of co-stimulatory pathways can be achieved through modifications in the intracellular domain of the CAR-T cells. Co-stimulatory domains (like via CD28, 4-1BB, CD27, or CD134) augment the durability and effectiveness of CAR-T cells by delivering supplementary activation cues and fostering T-cell proliferation. Another approach is a supplementary cytokine or immune modulator transgene, expressed either constitutively or upon CAR-T-cell activation, to regulate the tumor microenvironment by releasing cytokines, thereby bolstering T-cell activity and attracting other immune cells [311]. The current focus of research in the field of CAR-T-cell technology is the development of so-called “logic-gated” chimeric antigen receptors (CARs). There are various approaches for logic-gated CARs. AND logic-gate CARs require two specific antigens on tumor cells for activation. This approach could improve the specificity of tumor cell recognition. Furthermore, the approach of researching OR logic-gate CARs offers the possibility of addressing several tumor antigens (e.g., dual-CAR-T cells, tri-CAR-T cells, quad-CAR-T cells). This strategy proves advantageous in addressing the issue of tumor-associated antigen escape or antigen loss. Furthermore, by designing so-called inhibitory CARs (iCARs, NOT logic-gate CARs), off-target toxicity can be reduced. iCARs can recognize antigens that are expressed on normal and healthy tissues. Upon binding, inhibitory signals prevent an autoimmune reaction against the body’s own tissues [311]. Taken together, these novel technologies have the potential to increase the effectiveness of the therapy and at the same time reduce therapy-associated side effects.

The route of administration, and duration of action of CAR-T cells pose further challenges in the therapy of HCC. The administration of CAR-T cells leads to T-cell depletion. T cells lose the ability to proliferate, secrete cytokines, and destroy tumor cells during prolonged antigen stimulation. One cause of this is the compensatory overexpression of immunosuppressive receptors such as PD-1, TIM-3, and LAG-3. This process significantly limits the anti-tumor effect. Combining CAR-T cells with the use of immune checkpoint inhibitors thus presents a potential therapeutic approach. Furthermore, it is known that the tumor environment in patients is often characterized by fibrotic remodeling. This significantly impairs the migration and infiltration capabilities of CAR-T cells towards the tumor after intravenous infusion. To overcome this hurdle, CAR-T cells with chemokine receptors and heparinase-expressing CAR-T cells have already been developed. Another possibility to overcome the problem pf CAR-T-cell migration to the tumor is the intratumoral administration of CAR-T cells [311].

The potential of different CAR-T cells, including combination strategies and novel designed CARs, is under investigation in several clinical trials (see Table 19).

### 4.16. Therapeutic Vaccination Strategies in the Setting of HCC

Therapeutic vaccination includes various therapeutic approaches, including oncolytic viruses, vaccination with tumor antigens, and administration of immune cells induced with tumor antigens (dendritic cell therapy).

Vaccination using AFP-derived peptides demonstrated a manageable safety profile and clinically meaningful anti-tumor efficacy for HCC in a phase I study [312]. In this study, 15 patients were enrolled and received injections of 3 mg AFP-derived peptides (AFP357 and AFP403) every 2 weeks for at least 6 weeks. Among these patients, one patient showed a complete response, while tumor growth was also reported in eight patients.

Another phase I trial investigated GP3C-derived peptides in patients with advanced HCC, showing well-tolerated vaccination, measurable immune responses, and anti-tumor efficacy [313]. Additionally, GP3C-derived peptide vaccination significantly lowered the recurrence rate in the adjuvant setting after resection of HCC. In a phase II trial, 33 patients underwent surgery alone, while 35 patients underwent surgery plus GP3C-derived peptide vaccination, resulting in a one-year recurrence rate of 48% for surgery alone compared to 24% in the vaccination arm [314].

For peptides derived from multidrug resistance-associated protein 3 (MRP3) and telomerase reverse transcriptase (TERT), early phase I clinical trials reported induction of vaccine-specific T cells and a tolerable safety profile [315,316].

However, in a single-arm phase II clinical trial, the combination of low-dose cyclophosphamide and the telomerase peptide (GV1001) vaccine in patients with advanced HCC demonstrated no anti-tumor efficacy [317].

In the HepaVac-101 trial (phase I/II, NCT03203005), a multi-peptide-based HCC vaccine (IMA970A) combined with the TLR7/8/RIG I agonist CV8102 was assessed for the adjuvant setting in patients with HCC in BCLC stages A and B. The trial induced immune responses against ≥1 vaccinated HLA class I tumor-associated peptide in 37% and ≥1 vaccinated HLA class II tumor-associated peptide in 53% of the vaccinees [318].

However, therapy with peptide vaccines poses challenges due to the stimulation of autoreactive T cells and potential side effects and organ damage resulting from the expression of these peptides on normal cells.

Dendritic cells (DCs) are the primary antigen-presenting cells crucial for activating cytotoxic T cells in the context of immunological anti-tumor response. To create DC vaccines, autologous monocytes are typically collected and stimulated using specific cytokines (GM-CSF and IL-4). These cells are then loaded with tumor lysates, tumor-derived proteins and peptides, and nucleic acids (i.e., DNA and RNA) and reintroduced into patients [319]. Several phase I/II trials have reported safety data, indications of anti-tumor efficacy, and immune responses from patients with HCC vaccinated using monocyte-derived DCs loaded with AFP, MAGE A-1, GPC-3, or pulsed with tumor lysate [320,321,322,323,324,325,326].

In a clinical phase I trial (NCT01974661), the efficacy and safety of the immune primer ilixadencel (pro-inflammatory allogeneic dendritic cells) were compared in combination with sorafenib and as monotherapy, and 11 patients (73%) showed an increased frequency of tumor-specific CD8+ T cells in peripheral blood and 1 patient showed a partial response (with ilixadencel as monotherapy). Additionally, stable disease was observed as the overall best response in five patients. OS ranged widely from 1.6 to 21.4 months. The authors concluded that monotherapy with ilixadencel and its combination with sorafenib demonstrated a meaningful safety profile [327].

Further randomized clinical studies are necessary to evaluate the efficacy of DC vaccines in the treatment of HCC. The lack of an impressive tumor response in the previous studies may be attributed to several factors. Insufficient functionality of ex vivo-induced dendritic cells, inadequate migration in the lymphoid tissue around the tumor, and the immunosuppressive tumor microenvironment are among the discussed factors. Current studies aim to combine dendritic cell-based vaccination therapy with CPI usage to overcome these challenges [328].

Oncolytic viruses are designed to replicate within cancer cells, causing their destruction (oncolysis). JX-594 (Pexa-Vec) is among the most used oncolytic virus in clinical trials for HCC. A phase II clinical trial (NCT00554372) demonstrated oncolytic efficacy and tumor responses in patients with advanced HCC [329].

However, the TRAVERSE trial (phase II, NCT01387555), comparing treatment with JX-594 plus best supportive care (BSC) versus BSC alone in 129 patients with advanced HCC after failure of treatment with sorafenib, showed that JX-594 did not improve overall survival, ORR, time to progression, or safety compared to BSC alone [330].

Another phase III randomized trial (PHOCUS, NCT02562755) comparing oncolytic therapy with JX-594 followed by sorafenib versus sorafenib alone in patients with advanced HCC without prior systemic therapy was terminated. The interim results suggested an unlikelihood of meeting the primary objective by the final analysis. The planned study exploring JX-594 in combination with nivolumab for first-line treatment in advanced HCC (phase I/II trial NCT03071094) was halted due to negative outcomes from the phase III trials PHOCUS and CheckMate 459. Further investigations are required to explore the potential of JX-594, particularly in combination with established immunotherapies, for first-line treatment in HCC.

One reason for the limited efficacy of cancer vaccines may lie in immunological tolerance to tumor-associated antigens, which inhibits the induction of an anti-tumor immune response. Unlike tumor-associated antigens, neo-antigens are absent from healthy cells and differ from germline antigens, rendering them ideal targets for anti-tumor vaccine treatments. Identifying these antigens presents a future challenge. Theoretical advantages of research on these vaccine types include fewer side effects due to the tumor specificity of neo-antigens and the potential for improved long-term tumor control through the induction of memory T cells [331]. The immunosuppressive microenvironment found in liver cancer presents a challenge to the effectiveness of cancer vaccines. Combining these vaccines with other immunotherapies like immune checkpoint inhibitors will likely be necessary to enhance their efficacy. The potential for combining vaccines with other treatments, like combining them with chemoembolization/tyrosine kinase inhibitors or local ablative therapy, is under investigation in several clinical trials (see Table 20).

### 4.17. Cytokine-Directed Therapeutic Regimens in the Setting of HCC

Cytokines play a crucial role in regulating the immune system, contributing to tumor control. In recent years, clinical research has increasingly focused on utilizing cytokines in tumor therapy. There is also ongoing investigation into agonists targeting co-stimulatory checkpoint pathways in early clinical trials (see Table 21).

TNF alpha has shown promising results in reducing the early recurrence of HCC after resection, as observed in a meta-analysis [332,333]. Additionally, a meta-analysis revealed that pegylated interferon reduced recurrence rates and improved overall survival in patients who underwent liver resection for HCC due to hepatitis B or C infections [334].

Transforming growth factor β (TGF-β), a cytokine belonging to the TGF-β superfamily, comprises three isoforms (TGF-β 1-3). TGF-β plays a role in cancer progression. Initially, in the early stages of tumor development, TGF-β acts as a tumor suppressor by exhibiting antiproliferative, proapoptotic, and antiangiogenic effects. However, in the process of cancer developing, the response of cancer cells to TGF-β changes. They no longer undergo cell cycle arrest and apoptosis. Instead, TGF-β promotes cancer progression by shielding cancer cells from the immune system through its immunosuppressive effects [335,336].

Galunisertib, an inhibitor of TGF-β1 receptor type I, was investigated in a phase II trial in combination with sorafenib. In the dose expansion cohort, the 47 patients with advanced HCC, without prior systemic treatment, received 150 mg of galunisertib twice daily alongside 400 mg of sorafenib twice daily. The mOS was 18.8 months. Two patients achieved a partial response, while twenty-one showed stable disease. Interestingly, the trial observed sequential determination of TGF-β levels, revealing that patients with declining serum levels during therapy had significantly prolonged mOS compared to non-responders [337]. Galunisertib was also studied in a phase II trial (NCT01246986) as a second-line monotherapy for patients after sorafenib treatment failure. The trial enrolled 149 patients divided into two groups: Part A (AFP ≥ 1.5× ULN) or Part B (AFP < 1.5× ULN). The group with low-AFP tumors demonstrated a significant increased mOS of 16.8 months (95% CI; 10.5–24.4) compared to the AFP-high cohort, which showed an mOS of 7.3 months (95% CI; 4.9–10.5). Interestingly, responders to TGF-β1 and high-AFP expression exhibited longer OS compared to non-responders [338].

Nivolumab in combination with galunisertib was investigated in a phase I/II trial (NCT02423343). The phase I cohort, encompassing several solid tumors, demonstrated a manageable safety profile. However, the phase II cohort, which included only one patient with HCC, showed a PFS of 5.4 months and an OS of 14.5 months. Hence, comprehensive efficacy results for the use of nivolumab and galunisertib in HCC patients are yet to be determined [339].

Endoglin, an integral transmembrane glycoprotein and coreceptor for TGF-β ligands, was the target of carotuximab, an anti-endoglin antibody. In a phase II trial (NCT01375569) involving patients with advanced HCC after sorafenib treatment failure, carotuximab as a single agent did not demonstrate anti-tumor efficacy [340]. However, a subsequent phase I/II trial (NCT01306058) evaluated carotuximab in combination with sorafenib. The dose escalation phase established a recommended dose of 15 mg/kg carotuximab every 2 weeks for the phase II cohort. Among 25 treated patients, an ORR of 21% was observed. The mPFS was 3.8 months (95% CI; 3.2–5.6), and the mOS was 15.5 months (95% CI; 8.5–26.3) [341]. Further randomized studies are required to clarify the role of TGF-β and endoglin inhibitors in HCC treatment.

Recently, data from a phase I study (NCT02315066) of ivuxolimab, an OX40 agonist, were reported. This study included patients with advanced solid tumors, including 19 patients with HCC, and showed a manageable safety profile along with preliminary anti-tumor activity [342].

Cixutumumab, a monoclonal antibody targeting the type I IGF receptor, demonstrated only limited clinical efficacy in unselected treatment-naive patients with advanced HCC. In a phase II trial (NCT00906373), the combination of cixutumumab plus sorafenib resulted in an mPFS of 6.0 months and an mOS of 10.5 months [343]. Cixutumumab monotherapy also failed to show clinically meaningful activity in unselected patients with HCC (phase II, NCT00639509) [344].

## 5. Adjuvant Treatment of HCC

Adjuvant treatment for HCC refers to therapies after surgery or local ablation to reduce the risk of cancer recurrence. Approaches under investigation in clinical trials include targeted therapy and immunotherapy. Adjuvant therapy is tailored to the individual patient based on factors such as the stage of the cancer, underlying liver function, histopathological risk factors, and overall health. Clinical trials are ongoing to evaluate the effectiveness of different adjuvant treatments in improving outcomes for patients with HCC (see Table 22).

Besides the approval of TKIs in the setting of advanced tumor status, there is only limited or negative data on the use of TKIs in the adjuvant setting following resection or liver transplantation. The role of sorafenib as adjuvant treatment post-resection or after local ablation was investigated in the STORM trial (phase III, NCT00692770). Unfortunately, adjuvant treatment with sorafenib did not demonstrate a reduction in postoperative tumor recurrence compared to placebo in the STORM trial. The study reported an overall short duration of therapy (median treatment duration was 12.5 months, despite permitting 4 years of therapy) and frequent dose modifications (reported in 89% of cases). Additionally, the trial included patients at low risk of relapse, and the median recurrence-free survival did not show significant differences between the two groups [345]. In contrast, two small retrospective studies conducted in China revealed that adjuvant sorafenib following liver resection notably enhanced both DFS and OS among patients with BCLC C-stage HCC [346,347]. Furthermore, a recent meta-analysis comprising nine studies also concluded that sorafenib holds value as adjuvant therapy [348]. Additionally, ongoing studies are exploring the potential of lenvatinib in the adjuvant setting post-resection or after liver transplantation and for tumor recurrence after liver transplantation. A retrospective study demonstrated an adjuvant effect of lenvatinib following liver resection in patients with macrovascular invasion and HBV-related HCC, which extended both disease-free survival and overall survival [349]. Another retrospective study demonstrated efficacy in adjuvant treatment for patients with a high residual alpha-fetoprotein level after resection or ablation [350].

There is now sound clinical evidence for the use of immunotherapy in the adjuvant setting. The efficacy and safety of adjuvant nivolumab after liver resection or ablation was investigated in a single-arm prospective study in patients with HCC and after liver resection, with a median DFS of 26.3 months [351]. Another study on the use of PD-1 inhibitors as adjuvant therapy demonstrated a significantly better DFS compared to the control group [352]. Nevertheless, a crucial risk of adjuvant treatment with immune checkpoint inhibitors is transplant rejection or reactivation of hepatitis B in HBV-associated HCC. Recently, the IMbrave050 trial investigated the use of atezolizumab and bevacizumab in the adjuvant setting following curative resection or ablation. Patients were randomly assigned to receive atezolizumab plus bevacizumab for 1 year or undergo active surveillance. The primary endpoint was recurrence-free survival. Recently published data demonstrated a notable 28% improvement in recurrence-free survival for patients treated with atezolizumab plus bevacizumab [353]. The IMbrve050 study included patients at high risk for tumor recurrence after resection or ablation based on criteria such as tumor size and number, as well as histopathological criteria (lymphatic and blood vessel invasion, grading). Despite the positive data, there is currently no approval or implementation in international guidelines. In the IMbrave050 study, patients with a history of liver transplantation were excluded, leaving the significance of adjuvant therapy in this setting unanswered. However, it is worth noting that 63% of the patients had HBV-associated HCC, although the incidence of HBV reactivation has not been reported thus far [354].

There is also evidence for efficacy of TACE and HAIC in the adjuvant setting. Two controlled randomized clinical trials were able to demonstrate a benefit for the use of TACE in the adjuvant setting compared to sole tumor surveillance [355,356]. A meta-analysis comprising 40 studies (10 RCTs/30 non-RCTs) demonstrated that patients with high risk factors for tumor-recurrence benefit from adjuvant TACE, exhibiting longer overall survival and disease-free survival. However, conversely, this study found no clinical benefit patients without macrovascular invasion [357]. Notably, the Chinese guidelines for the treatment of HCC recommend the use of TACE in the adjuvant setting. Prospective randomized studies regarding the use of HAIC in the adjuvant setting demonstrated controversial results. Three RCTs were able to demonstrate a benefit in terms of overall survival and disease-free survival, while another study only showed a marginal benefit of adjuvant HAIC [358,359,360,361]. A meta-analysis affirmed the efficacy of postoperative adjuvant HAIC in enhancing prognosis and indicated that patients with macrovascular invasion and portal vein tumor thrombus, especially benefit from adjuvant treatment [362]. Based on the existing evidence, combinations of systemic therapy and TACE or HAIC are currently under extensive investigation (see Table 22).

There are also data on intensifying adjuvant treatment by combining local and systemic treatment approaches. The combination appears attractive from various perspectives. On the one hand, micro-metastases, lymph node infiltration, and tumor cells that spread intraoperatively and also have local effects due to residual tumor manifestations at the resection margin can be addressed. The combination of TACE plus lenvatinib was investigated in a multicenter prospective cohort study that screened patients with high risk factors for tumor recurrence. The authors demonstrated an impressive improvement in the median disease-free survival of 17.0 months ((95% CI; 12.0–24.0) in the TACE plus lenvatinib group, compared to the TACE group, with a median disease-free survival of 9 months (95% CI; 7.0–14.0, *p* = 0.0228; HR 0.6, 95% CI; 0.4–1.0)) [363]. Therefore, several clinical trials exploring the effectiveness of TACE combined with systemic therapy to prevent tumor recurrence are ongoing (see Table 22).

There is also evidence supporting the use of therapeutic vaccination and the use of cytokine-induced killer cells in the adjuvant setting. For details on the existing scientific data from previous studies, we kindly refer to Section 4.15 and Section 4.16. Regarding these therapeutic strategies, there are currently no recommendations in international guidelines, and from our perspective, further clinically randomized studies are necessary to assess their significance.

## 6. Conclusions

Oncologic systemic therapy represents the standard of care in the treatment of advanced HCC at BCLC stage C. With the update of the BCLC classification in 2022, systemic therapy is already considered for patients with intermediate-stage HCC (BCLC stage B) through the concept of treatment stage migration (TSM). Established standard treatment in first-line therapy is a combination of immunocheckpoint inhibitors (double immunocheckpoint inhibition) or a combination of an immunocheckpoint inhibitor and a VEGF-directed antibody. In the second-line setting or in the case of contraindications regarding immunocheckpoint blockade, the known TKIs are available. The currently approved drugs are listed in Table 23. A timeline of approved and currently recommended treatment options for HCC in the palliative setting in international guidelines is shown in Figure 4.

In general, the prognosis of HCC remains poor, particularly in advanced stages. However, advancements in early detection, treatment modalities, and personalized medicine are improving outcomes and offering hope for better prognosis in certain patient populations. The future of HCC treatment may lie in combination therapies that target multiple pathways involved in tumor growth and progression.

This study showed the enormous progress in the development of molecular therapy approaches for HCC. Numerous clinical trials are underway. The use of combination therapies will determine the future treatment of HCC, as synergistic mechanisms of action and the potential of overcoming therapy resistance of different molecular therapy approaches can be utilized. This includes combinations of immunotherapy with targeted therapies and locoregional therapies. However, as therapy approaches are intensified, therapy-associated toxicity generally increases. There are limitations here, particularly in patients with HCC and underlying liver cirrhosis.

Results from the phase I/II MORPHEUS-Liver study favors the triple therapy with atezolizumab/bevacizumab and the anti-TIGIT antibody tiragolumab. With an impressive overall response rate the triplet of tiragolumab plus atezolizumab/bevacizumab may represent a novel first-line treatment option for advanced HCC. The ongoing phase III study IMbrave152/SKYSCRAPER-14 will further investigate this triplet combination in a first-line setting.

With the recent update of the BCLC classification, oncological systemic therapy for patients with HCC is increasingly moving into earlier tumor stages. For example, the concept of TSM now allows the use of systemic therapy in patients with stage BCLC B. Further investigations into the best procedure and the most effective therapy sequence in the event of a successful tumor response will be particularly interesting. The question of whether patients benefit from sequential local ablative treatment (ablation, resection), TACE, or even liver transplantation after successful downstaging using systemic therapy is currently unanswered. There are currently interesting study approaches that address this issue and should provide further evidence in the future to clarify this question.

Another option for future therapy will be the combination of systemic therapy and TACE/local ablation. Combining TACE with systemic therapy may have synergistic effects. TACE induces tumor necrosis, releasing tumor antigens and creating an inflammatory response, which could potentially enhance the efficacy of systemic therapies, particularly immunotherapy. Systemic therapy, on the other hand, can help in addressing micro-metastases or residual disease that TACE might not effectively target alone. Positive data has already been presented with the EMERALD-1 study.

For adjuvant treatment following resection or ablation in early tumor stages, there is strong clinical evidence from the IMbrave050 study. High-risk patients with hepatocellular carcinoma demonstrated significantly prolonged recurrence-free survival following curative resection or ablation with the combination of atezolizumab and bevacizumab compared to active observation. Approval in the adjuvant setting and inclusion in guidelines is anticipated. In our view, adjuvant therapy approaches will become increasingly important in the treatment of HCC in the future, as there is a comparatively high risk of recurrence even after curative treatment. Adjuvant treatment has become firmly established in other tumor entities of the gastrointestinal tract due to significant progress in terms of prolonging the survival of patients. We anticipate that adjuvant treatment of patients with HCC will be included in international guidelines soon, particularly for patients with a high risk of tumor recurrence.

Several new molecular pathways involved in tumor cell growth (cell differentiation, cell proliferation, apoptosis), angioneogenesis, and forming the tumor microenvironment are under clinical development, including cell-derived immunotherapy and vaccination. Advances in understanding of these molecular pathways have paved the way for the emergence of targeted therapies. Numerous promising novel anticancer agents are currently being studied for HCC treatment, with ongoing clinical trials showing potential to enhance anti-tumor efficacy in patients with advanced HCC. This includes investigations into the use of adoptive-cell therapy (ACT), CART cells and cytokine-directed therapeutic regimens, as well as addressing DNA damage response repair. Thus, we emphasize bioptic confirmation also to acquire human biological tissue samples for further research.

## Figures and Tables

**Figure 1 cancers-16-01831-f001:**
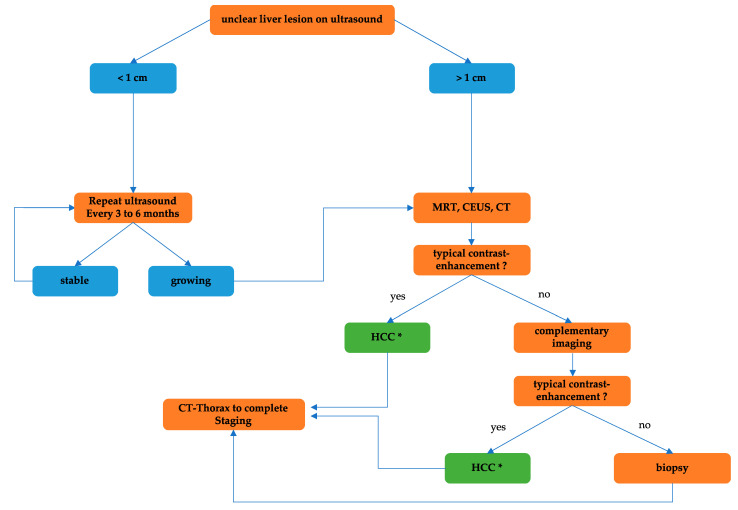
Diagnostic algorithm for hepatocellular carcinoma modified to EASL guideline (European Association for the Study of the Liver), NCCN guideline (National Comprehensive Cancer Network) and German S3-Leitlinie. * Biopsy is recommended in case of palliative setting before initiating systemic therapy.

**Figure 3 cancers-16-01831-f003:**
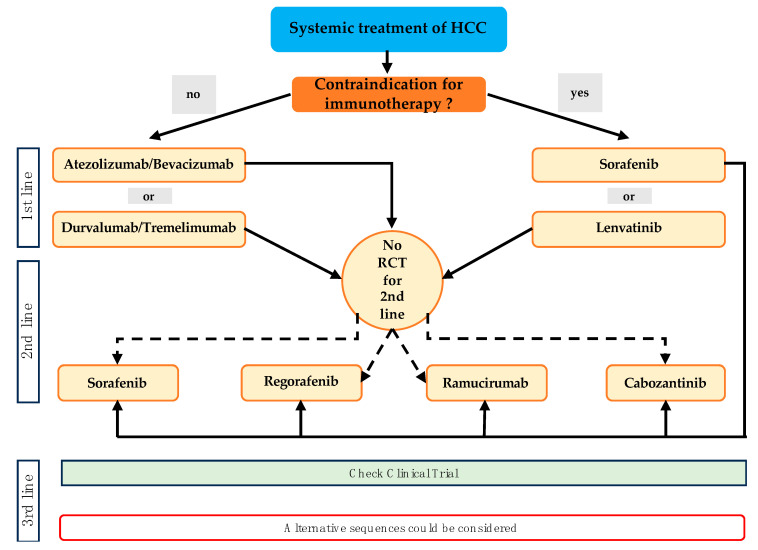
Systemic treatment of HCC recommended by international guidelines. RCT: randomized controlled trial. Dashed lines: no RCTs are available, use to be considered on the basis of historical data from pivotal studies.

**Figure 4 cancers-16-01831-f004:**
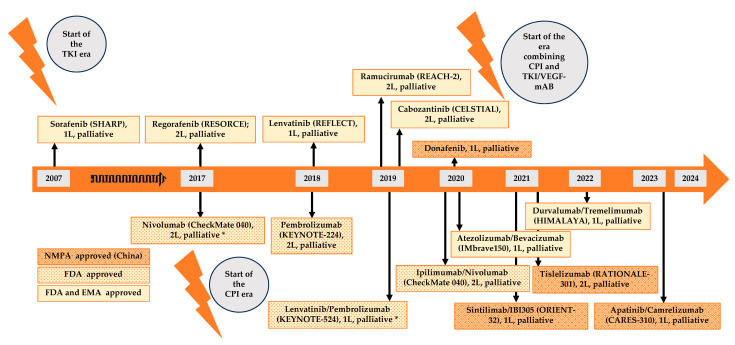
Timeline of milestones in targeted therapy with approval for treatment of advanced hepatocellular carcinoma. 1L: first-line; 2L: second-line; CPI: immune checkpoint inhibitor; TKI: tyrosine-kinase inhibitor; VEGF-mAb: monoclonal antibodies targeting the vascular endothelial growth factor pathway. * Approval withdrawn.

**Table 1 cancers-16-01831-t001:** Risk factors for HCC [14].

Risk-Factor	Study-Design	Risk Estimate (95% CI)	Reference
acute intermittent porphyria	case-control	8.0 (24.3–59.3) *; 5.15 (2.90–9.12)	[15,16,17,18]
aflatoxin	case-control	6.37 (3.74–10.86)	[19,20,21]
alcohol abuse	case-control	7.3 (6.8–7.8)	[22,23]
α1-antitrypsin deficiency	case-control	5 (N/A); 3.91 (2.67–5.74)	[18,24,25,26,27]
biliary cirrhosis	case-control	12.28 (8.86–17.02)	[18]
chronic hepatitis B viral infection	case-control	21.6 (17.9–26.0); 7.65 (4.6–24.95)	[23,28,29,30,31,32,33]
chronic hepatitis C viral infection	case-control	59.9 (55.1–65.1); 109.88 (71.12–207.10)	[23,28,31,32,33,34,35]
cirrhosis of the liver	case-control	808.37 (257.52–2537.52)	[33,36,37]
diabetes mellitus	case-control	3.00 (2.75–4.41)	[33,38,39,40]
Gaucher disease	N/A	N/A	[41,42]
glycogen storage disease	N/A	N/A	[43]
hemochromatosis	case-control	10.89 (4.38–27.09)	[18,44]
non-alcoholic fatty liver disease	case-control	7.55 (4.89–11.26)	[33,45,46,47]
tyrosinemia type I	N/A	N/A	[48,49]
Wilson disease	case-control	4.1 (1.80–9.33)	[18]

* primary liver cancer (including HCC and cholangiocarcinoma); N/A: not available.

**Table 2 cancers-16-01831-t002:** Development of mOS and mPFS in clinical trials with sorafenib for advanced HCC.

Study	Year	Phase	mOS	mPFS	NCT
SHARP	2008	III	10.7	-	NCT00105443
BRSIK-FL	2013	III	9.9	-	NCT00858871
SUN1170	2013	III	10.2	3.0	NCT00699374
LiGHT	2015	III	9.8	-	NCT01009593
REFLECT	2018	III	12.3	3.7	NCT01761266
IMbrave150	2020	III	13.4	4.3	NCT03434379
CheckMate 459	2022	III	14.7	-	NCT02576509
COSMIC-312	2022	III	15.5	4.2	NCT03755791
HIMALAYA	2022	III	13.8	4.1	NCT03298451
RATIONALE-301	2023	III	14.1	3.4	NCT03412773
CARES-310	2023	III	15.2	3.7	NCT03764293

**Table 3 cancers-16-01831-t003:** Ongoing studies for sorafenib in HCC.

Study	Phase	Status	NCT
Predictors of sorafenib response in patients with advanced HCC	IV	Not yet recruiting	NCT05967429
Real-world study of efficacy and safety of ICIs and TKIs therapy for HCC	IV	Active, recruiting	NCT05420922

**Table 4 cancers-16-01831-t004:** Ongoing studies for HAIC in HCC.

Study	Phase	Setting	Treatment Arm A	Treatment Arm B	Primary Endpoint	NCT
-	II	1L, palliative + PVTT	HAIC plus Tor	HAIC + Sor	PFS	NCT04135690
-	III	1L, palliative	HAIC + Sor	TACE + Sor	OS	NCT02856126
HAICPD1-HCC	II	neoadjuvant	HAIC + Sin	HAIC	PFS	NCT03869034
-	II	Palliative	HAIC + Dur	-	OS	NCT04945720
-	II	1L, palliative	HAIC + Sin + B	-	ORR	NCT05029973
-	II	neoadjuvant	HAIC + Dur + Tre + B	-	ORR	NCT05864755
-	II	1L, palliative	HAIC + Len + Tis	-	ORR	NCT05954897
-	II	1L, palliative	HAIC + Len + PD-1	Len + PD-1-	PFS	NCT05166239
-	II	perioperative	Sin + Len	HAIC	DFS	NCT05519410
-	II	Palliative	HAIC + Sin + B	-	PFS	NCT05617430
-	II	neoadjuvant	HAIC + Cam + Apa	-	R0-rate; SCR	NCT05099848
-	II	Palliative	HAIC+ Cam + Len or Rego	-	PFS	NCT05135364
D-TRIPLET	II	Palliative	HAIC + B + Sin	-	ORR	NCT05214339
-	III	Palliative	HAIC + Len + PD-1	HAIC (Len + PD-1 sequential)	OS	NCT06041477
-	III	Palliative	HAIC + Cam + Apa	Cam + Apa	OS	NCT05198609
TRIPLET	II	Palliative	HAIC + Cam + Apa	-	ORR	NCT04191889
-	II	1L, palliative	HAIC + A + B	-	PFS	NCT05886465
TRIPLET-III	III	1L, palliative	HAIC + Cam + Apa	Cam + Apa	PFS	NCT05313282
-	II	Palliative	HAIC + Len + Cam	-	ORR	NCT05003700

1L: first-line; A: atezolizumab; Apa: apatinib; B: bevacizumab; Cam: camrelizumab; Dur: durvalumab; Len: lenvatinib; Rego: regorafenib; Tis: tislelizumab; Tor: toripalimab; Sin: sintilimab; TACE: transcatheter arterial chemoembolization; PFS: progression-free survival; DFS: disease-free survival; OS: overall survival; ORR: overall response rate; SCR: surgical conversion rate; PD-1: programmed cell death protein 1.

**Table 5 cancers-16-01831-t005:** Ongoing studies for lenvatinib in HCC.

Study	Phase	Setting	Treatment Arm A	Treatment Arm B	Primary Endpoint	NCT
STELLAR	IV	palliative	Len or Sor	-	Safety	NCT04763408
-	IV	1L palliative	Len	-	Safety	NCT04297254
-	II	recurrence after LTx	Len	-	ORR	NCT05103904

1L: first-line; Len: lenvatinib; LTx: liver transplantation; ORR: overall response rate; Sor: sorafenib.

**Table 6 cancers-16-01831-t006:** Ongoing studies for regorafenib/cabozantinib in HCC.

Study	Phase	Setting	Treatment Arm A	Treatment Arm B	Primary Endpoint	NCT
REGONEXT	II	2L, palliative (after A/B)	Rego	-	PFS	NCT05134532
CaPture	II	2L, palliative, refractory to PD-1	Cabo	-	ToT	NCT04767906
AURORA	II	2L, palliative (after Len or Len + IO)	Cabo	-	ToT	NCT04511455
-	I/II	2L, palliative, CP B	Cabo	-	MTD, RP2D	NCT04497038

2L: second-line; A: atezolizumab; B: bevacizumab; Cabo: cabozantinib; CP: Child–Pugh; MTD: maximum tolerated dose; Rego: regorafenib; IO: immunotherapy; RP2D: recommended phase 2 dose; ToT: time on treatment.

**Table 7 cancers-16-01831-t007:** Randomized studies for different TKIs in HCC with negative outcomes.

Study	Phase	Treatment	mOS Experimental	mOS Control	NCT
BRISK-FL	III	brivanib vs. sorafenib	9.5 months	9.9 months	NCT00858871
LiGHT	III	linifanib vs. sorafenib	9.1 months	9.8 months	NCT01009593
SUN1170	III	sunitinib vs. sorafenib	7.9 months	10.2 months	NCT00699374
-	II	dovotinib vs. sorafenib	8.0 months	8.4 months	-
-	I/II	tivozanib	9.0 months	-	NCT01835223
-	II	cediranib	5.8 months	-	NCT00238394

**Table 8 cancers-16-01831-t008:** Ongoing studies for CPIs in HCC (without TKI combination).

Study	Phase	Setting	Treatment Arm A	Treatment Arm B	Primary Endpoint	NCT
CheckMate 9DW	III	1L, palliative	Ipi plus Nivo	Sor/Len	OS	NCT04039607
PRIME-HCC	I/II	neoadjuvant	Ipi plus Nivo	-	safety + delay to surgery	NCT03682276
-	II	2L, palliative (after A/B)	Ipi plus Nivo	-	ORR	NCT05199285
NEOTOMA	II	neoadjuvant/adjuvant	Dur plus Tre	-	safety	NCT05440864
SIERRA	III	1L, palliative	Dur plus Tre	-	safety, ORR	NCT05883644
-	II	neoadjuvant before liver transplantation	Dur plus Tre	-	cellular rejection rates	NCT05027425
-	I	neoadjuvant (resection)	Nivo	Nivo + Rela	complete pre-op treatment and proceed to surgery	NCT04658147
-	III	1L, palliative	Cam + FOLFOX4	Cam	OS	NCT03605706
ACROPOLI	II	palliative	Spartalizumab	Tis	ORR	NCT04802876
-	II	palliative	INCB086550	-	ORR	NCT04629339
-	I	palliative	MT-8421 plus Nivolumab	MT-8421	Safety, ORR	NCT06034860
-	IV	palliative	Cam	-	safety	NCT04947956
-	III	1L, palliative	Cam plus FOLFOX4	FOLFOX4	OS	NCT03605706
-	I/II	1L, palliative	CHS-006 (anti-TIGIT) + Tor	-	safety	NCT05757492
-	II	1L, palliative	Nivo + Cabi	Nivo	ORR	NCT04050462

1L: first-line, 2L: second-line; A: atezolizumab; B: bevacizumab; Cabi: cabiralizumab; Cam: camrelizumab; Dur: durvalumab; Ipi: ipilimumab; Len: lenvatinib; Nivo: nivolumab; OS: overall survival; ORR: overall response rate; Rela: relatlimab; Sor: sorafenib; Tor: toripalimab; Tre: tremelimumab.

**Table 10 cancers-16-01831-t010:** Ongoing studies for combination of target VEGF and CPIs in HCC.

Study	Phase	Setting	Treatment Arm A	Treatment Arm B	Primary Endpoint	NCT
-	II	palliative	Sor + Nivo	-	ORR, MTD	NCT03439891
-	I/II	palliative	Sor + Pembro	-	ORR	NCT03211416
GOING	II	2L, palliative	Rego + Nivo	-	safety	NCT04170556
REGOMUNE	I/II	2L, palliative	Rego + Ave	-	safety	NCT03475953
-	II/III	1L, palliative	Fino + SCT510	Sor	OS, PFS	NCT04560894
PRIMER-1	II	perioperative	Pembro	Lenv (Arm C: Len + Pembro)	MPR	NCT05185739
-	II/III	1L, palliative	Rulonilimab + Len	Len	ORR	NCT05408221
-	I/II	1L, palliative	Len + Cam	-	ORR	NCT04443309
-	II	1L, palliative	Len + Tori	-	ORR	NCT04368078
-	III	1L, palliative	Len + Tori	Len	OS	NCT04523493
-	I/II	1L, palliative	Cadonilimab + Len	-	ORR	NCT04444167
-	II	2L, palliative	Candolilimab + B	-	DCR	NCT05760599
-	II	1L, palliative	Sinti + Len	-	ORR	NCT04042805
-	II	1L, palliative	Dur + Len	-	ORR	NCT05312216
Dulect2020-1	-	1L, palliative or prä LTx	Dur + Len	-	PFS, RFS	NCT04443322
-	II	perioperative	Tis + Len	-	safety	NCT04834986
-	II	palliative	A + Cabo or Len	Cabo or Len	OS, PFS	NCT05168163
PLENTY202001	-	prä LTx	Len + Pembro	-	RFS	NCT04425226
TALENT	II	neoadjuvant	Tis + Len	Tis	DFS	NCT04615143
-	II	1L, palliative	Tis + Rego	Rego	Safety, ORR, PFS	NCT04183088
-	II	neoadjuvant	Dur + Rego	-	ORR	NCT05194293
-		2L, palliative	PD-1 + Rego	-	PFS	NCT05048017
-	II	1L, palliative	Tori + B	-	Safety, ORR	NCT04605796
IMbrave152/ SKYSCRAPER-14	III	1L, palliative	Tira + A + B	A + B	PFS, OS	NCT05904886
-	II	2L, palliative	Rego + Pembro	-	ORR	NCT04696055
TRIPLET	II/III	1L, palliative	A + B	A + B + Ipi	ORR, OS	NCT05665348
AB7 Trial	II	1L, palliative Child–Pugh B	A + B	-	safety	NCT04829383
-	II	neoadjuvant	A + B	-	Safety, pCR	NCT04721132
MONTBLANC	II	1L, palliative	Dur + Tre and B in case of progress	Dur + Tre + B	ORR	NCT05844046
CAPT	II	neoadjuvant	Cam + Apa	Cam	RFR	NCT04930315
-	II	1L, palliative	Cam plus Apa plus mFOLFOX7	-	ORR	NCT05412589
-	I/II	1L, palliative	Adebrelimab plus Cam plus Apa	-	Safety, ORR	NCT05924997
-	II	neoadjuvant	Cam plus Apa plus Oxaliplatin	-	MPR	NCT04850040
-	II	Perioperative	Cam plus Apa	-	ORR	NCT04701060
-	II	1L, palliative	Dona + Sinti	-	PFS	NCT05162352
-	III	1L, palliative	Sinti + IBI310	Sor	OS, ORR	NCT04720716
-	I/II	1L, palliative	Dona + Tori	-	Safety, ORR	NCT04503902
-	II	neoadjuvant	Tis	Tis + Len	MPR	NCT05807776
-	II	palliative	Pembro + Quavonlimab + Len	-	Safety, ORR	NCT04740307

1L: first line; 2L: second line; A: atezolizumab; Apa: apatinib; Ave: avelumab; B: bevacizumab; Cabo: cabozantinib; Cam: camrelizumab; DCR: disease control rate; DFS: disease-free survival; Dona: donafenib; Dur: durvalumab; Fino: finotonlimab; Ipi: ipilimumab; Len: lenvatinib; LTx: liver transplantation; MPR: major pathological response; MTD: maximum tolerated dose; Nivo: nivolumab; ORR: overall response rate; pCR: pathologic complete response; Pembro: pembrolizumab; PFS: progression-free survival; Rego: regorafenib; RFR: recurrence-free rate; RFS: recurrence-free survival; Sinti: sintilimab; Sor: sorafenib; Tira: tiragolumab; Tis: tislelizumab; Tori: toripalimab; Tre: tremelimumab.

**Table 11 cancers-16-01831-t011:** Ongoing studies for combination of target VEGF and/or CPIs with TACE in HCC.

Study	Phase	Setting	Treatment Arm A	Treatment Arm B	Primary Endpoint	NCT
CHECKMATE 74W	III	locoregional	Nivo + Ipi + TACE	Nivo + TACE (Arm C: TACE alone)	Safety	NCT04340193
LEAP-012	III	locoregional	Len + Pembro + TACE	TACE	PFS, OS	NCT04246177
TACE-3	II/III	locoregional	TACE	TACE + Nivo	OS, TTTP	NCT04268888
CHANCE2202	IV	locoregional	TACE + PD-1/PD-L1 + TKI or B	TACE	PFS, OS	NCT05332496
CHANCE2201	IV	locoregional	TACE + PD-1/PD-L1 + TKI or B	PD-1/PD-L1 + TKI or B	OS	NCT05332821
-	I	locoregional	PD-1 + Len + TACE	-	Resection rate	NCT04974281
EMERALD-3	III	locoregional	Tre + Dur+ Len +TACE	Tre + Dur +TACE (Arm C: TACE alone)	PFS	NCT05301842
-	I/II	1L, palliative	Len + Tis	Len + Tis + TACE	ORR	NCT05842317
-	III	1L, palliative	TACE + Len synchron	TACE + Len sequential	OS	NCT05220020
-	III	palliative	TACE + Cam + Apa	TACE	PFS	NCT05320692
-	III	palliative	TACE + Sinti + B	TACE + Len	OS	NCT05985798
-	II	palliative	TACE + Sinti	-	ORR	NCT04297280
TASK-02	II	palliative	TACE + Sinti + B	-	ORR	NCT04954794
-	III	palliative	TACE + Len + Sinti	TACE + Len	OS	NCT05608200
MORNING	II	neoadjuvant	TACE + Cadonilimab	-	MPR	NCT05578430
-	II	palliative	TACE + Sinti + Fru	-	PFS	NCT05971199
-	II	palliative	TACE + Tis + Sor	-	OS	NCT04992143
-	II	palliative	TACE + Ipi + Nivo + Cabo	-	PFS	NCT04472767
-	III	perioperative	TACE + Cam + Apa	TACE	RFS	NCT05613478
DEMAND	II	palliative	TACE + A +B synchron	TACE + A + B sequential	OS	NCT04224636
T-Double	II	palliative	TACE + Sinti + B	-	ORR	NCT04796025
-	II	-	Dur + Tre +B	TACE + Dur+ Tre + B	PFS	NCT03937830
-	II	Prä LTx	Don + TACE	-	DSR	NCT05576909
-	I	palliative	TACE + Don + Tori	-	safety	NCT04605185
-	II	palliative	TACE + Cam + Apa	-	ORR	NCT05550025
-	II	intermediate stage	TACE + Dur + Trem	-	ORR	NCT03638141
-	II	intermediate stage	TACE + A + B	-	safety	NCT05776875
ROSE	IV	palliative	TACE + Rego	Rego	OS	NCT05811481
LEN-TAC	III	palliative	TACE + Len + Cam	Len	OS	NCT05738616
-	II	palliative	TACE + Cam	-	TTP	NCT04652492

1L: first-line; A: atezolizumab; Apa: apatinib; B: bevacizumab; Cabo: cabozantinib; Don: donafenib; DSR: downstaging success rate; Dur: durvalumab; Fru: fruquintinib; Ipi: ipilimumab; Len: lenvatinib; LTx: liver transplantation; MPR: major pathological response; Nivo: nivolumab; OS: overall surviaval; ORR: overall response rate; Pembro: pembrolizumab; PFS: progression-free survival; Rego: regorafenib; RFS: recurrence-free survival; Sinti: sintilimab; Sor: sorafenib; TACE: transcatheter arterial chemoembolization; Tis: tislelizumab; Tori: toripalimab; Tre: tremelimumab; TTP: time to progression; TTTP: time to TACE progression.

**Table 12 cancers-16-01831-t012:** Ongoing studies for bispecific antibodies in HCC.

Study	Phase	Treatment	Target 1	Target 2	Primary Endpoint	NCT
-	II	AK104	PD-1	CTLA-4	ORR	NCT04728321
-	I/II	AK104	PD-1	CTLA-4	ORR	NCT04444167
DUET-4	I	Bavunalimab	LAG-3	CTLA-4	safety	NCT03849469

ORR: overall response rate.

**Table 13 cancers-16-01831-t013:** Ongoing studies for ADCs in HCC.

Study	Phase	Treatment	Target	Drug	Primary Endpoint	NCT
-	I	TORL-4-500	N/A	N/A	safety	NCT06005740
-	I	MGC018-02	B7-H3	Duocarmycin	safety	NCT05293496

N/A: not available.

**Table 14 cancers-16-01831-t014:** Ongoing studies for ROS1/ALK/MET alterations.

Study	Phase	Setting	Treatment	Primary Endpoint	NCT
MATCH	II	palliative	Crizotinib for patients with MET amplification, ALK translocation or ROS1 translocation/inversion	ORR	NCT02465060

**Table 15 cancers-16-01831-t015:** Ongoing studies for targeting PI3K/AKT/mTOR signaling pathway.

Study	Phase	Setting	Treatment	Primary Endpoint	NCT
-	II	palliative	Lenvatinib, Trametinib, Everolimus	safety	NCT04803318
MATCH	II	palliative	Taselisib (PI3K)	ORR	NCT02465060
MATCH	II	palliative	Copanlisib (PI3K)	ORR	NCT02465060
MATCH	II	palliative	Sapanisertib (mTOR)	ORR	NCT02465060
MATCH	II	palliative	Capivasertib (AKT)	ORR	NCT02465060
MATCH	II	palliative	Ipatasertib (AKT)	ORR	NCT02465060

**Table 16 cancers-16-01831-t016:** Ongoing studies for targeting DKK1.

Study	Phase	Setting	Treatment	Primary Endpoint	NCT
-	I/II	palliative	DKN-01 + Sorafenib	Safety, time to progression	NCT03645980

**Table 17 cancers-16-01831-t017:** Ongoing studies for NTRK alterations.

Study	Phase	Setting	Treatment	Primary Endpoint	NCT
MATCH	II	palliative	Larotrectinib for patients with NTRK fusion	ORR	NCT02465060

**Table 18 cancers-16-01831-t018:** Studies addressing treatment with CIKs in HCC.

Study	Phase	Treatment	Treatment Arm A	Treatment Arm B	Primary Endpoint	NCT
- ^1^	II	palliative	CIK	No treatment	RFS	NCT02856815
- ^2^	II	-	TACE + CIK	TACE + follow-up	PFS	NCT01821482
- ^1^	II/III	BCLC A/B	MWA + CIK	MWA	survival	NCT02851784
- ^2^	II	palliative	TACE + CIK	TACE	OS	NCT02487017
- ^2^	III	BCLC B	TACE + CIK	TACE	-	NCT02568748
- ^2^	III	BCLC C/B	CIK	Best supportive care	-	NCT02568748
- ^2^	I/II	palliative	CIK + PD-1 Inhibitor	-	PFS	NCT02886897
Active, recruiting	I	palliative	CIK	-	safety	NCT04282044

OS: overall survival; PFS: progression-free survival, RFS: recurrence-free survival. ^1^ Completed, results not reported; ^2^ status unknown.

**Table 19 cancers-16-01831-t019:** Studies addressing CAR-T-cell therapy in HCC.

Study	Phase	Setting	Treatment	Primary Endpoint	NCT
- ^2^	I	palliative	GPC3-targeted CAR-T Cell	safety	NCT04121273
- ^1^	I	palliative	GPC3-targeted CAR-T Cell	safety	NCT02905188
- ^1^	I	palliative	GPC3-targeted CAR-T Cell	safety	NCT03884751
-	I	palliative	GPC3-targeted CAR-T Cell	safety	NCT04951141
- ^2^	I/II	palliative	GPC3-targeted CAR-T Cell	safety	NCT03084380
- ^2^	I/II	palliative	GPC3-targeted CAR-T Cell applied by transarterial infusion	safety	NCT02715362
- ^2^	I/II	palliative	GPC3-targeted CAR-T Cell applied by intratumor injection	safety	NCT03130712
-	I	palliative	GPC3-targeted CAR-T Cell	safety	NCT03198546
-	-	palliative	GPC3-targeted CAR-T Cell	safety	NCT03302403
-	I	palliative	GPC3-targeted CAR-T Cell	safety	NCT04506983
-	-	palliative	GPC3-targeted CAR-T Cell	safety	NCT05620706
-	-	palliative	GPC3-targeted CAR-T Cell	safety	NCT05926726
-	I	palliative	GPC3-targeted CAR-T Cell	safety	NCT05344664
-	I	palliative	GPC3-targeted CAR-T Cell	safety	NCT05003895
-	I	palliative	GPC3-targeted CAR-T Cell	safety	NCT05783570
-	I	palliative	GPC3-targeted CAR-T Cell	safety	NCT05070156
-	I/II	palliative	GPC3-targeted CAR-T Cell	safety	NCT05120271
-	I	palliative	GPC3-targeted CAR-T Cell	safety	NCT05155189
-	I/II	palliative	GPC3-targeted CAR-T Cell	safety	NCT05652920
ATHENA	I/II	palliative	GPC3-targeted CAR-T Cell	safety	NCT06084884
- ^2^	I	palliative	c-Met/PD-L1 CAR-T Cell	OS	NCT03672305
- ^2^	I	palliative	NKG2D CAR-T cells	safety	NCT04550663
-	I	palliative	NKG2D CAR T-cells	safety	NCT05131763
- ^2^	I/II	palliative	multi-target CAR-T cells	safety	NCT03638206
- ^2^	I/II	palliative	multi-target CAR-T cells	safety	NCT03941626
- ^2^	I	palliative	CD147 CAR-T cells applied by hepatic artery infusion	safety	NCT03993743
- ^2^	I/II	palliative	MUC1 CAR-T cells	safety	NCT02587689
- ^2^	I/II	palliative	EpCAM CAR-T cells	safety	NCT03013712
- ^2^	-	palliative	EpCAM CAR-T cells	DCR	NCT02729493
-	I	palliative	EpCAM CAR-T cells	safety	NCT05028933
-	I/II	palliative	B7/H3 CAR-T cells	safety, ORR	NCT05323201

DCR: disease control rate; ORR: overall response rate; OS: overall survival; ^1^ Completed, results not reported; ^2^ status unknown.

**Table 20 cancers-16-01831-t020:** Studies addressing therapeutic vaccination strategies in HCC.

Study	Phase	Setting	Treatment Arm A	Treatment Arm B	Primary Endpoint	NCT
-	I	palliative	Nivo + Ipi + Peptide Vaccine against DNAJB1-PRKACA Fusion Kinase	-	Safety, immune response	NCT04248569
Fusion VAC22	I	palliative	A + Peptide Vaccine against DNAJB1-PRKACA Fusion Kinase	-	Safety, immune response	NCT05937295
-	I	palliative	Pembro + Peptide Vaccine against P53	-	Safety	NCT02432963
-	I/II	palliative	Peptide Vaccine against HER-2/neu	-	Safety	NCT04246671
-	I/II	palliative	GNOS-PV02 Personalized Neoantigen Vaccine + Pembro + Plasmid-encoded IL-12	-	Safety, immune response	NCT04251117
-	-	palliative	mRNA Vaccine (ABOR2014/IPM511)	-	Safety	NCT05981066
TERTIO	II	palliative	A + B + Anti-telomerase Vaccine	A + B	ORR	NCT05528952
PNeoVCA	I	palliative	Pembro + Neoantigen Peptide Vaccine	-	Safety	NCT05269381
-	II	adjuvant	Nivo + Neoantigen Dendritic Cell Vaccine	-	RFS, immune response	NCT04912765
- ^2^	I	BCLC B	MWA + Neoantigen-based Dendritic Cell Vaccine	MWA	safety	NCT03674073
- ^2^	I	adjuvant	Neoantigen-primed DC Vaccine	-	DFS	NCT04147078
- ^2^	II	-	Autologous Dendritic Cell Vaccine + Surgery or TACE or Len/Sor	Surgery or TACE or Len/Sor	PFS	NCT04317248
- ^2^	I	palliative	Oncolytic Virus M1 (M1-c6v1) + Apatinib + SHR-1210 (Anti-PD-1)	-	Safety	NCT04665362
-	II	palliative	VG161 (Oncolytic Virus)	VG161 + Nivo	Safety, ORR, PFS	NCT05223816
-	I	palliative	Synov1.1 (Oncolytic Virus)	-	Safety, response	NCT04612504
-	II	palliative	RP3 (Oncolytic Virus) + A + B	-	ORR	NCT05733598
-	I	palliative	Oncorine (Oncolytic Virus) + Len + Tis	-	safety	NCT05675462

A: atezolizumab; B: bevacizumab; DFS: disease-free survival; Len: lenvatinib; MWA: microwave ablation; Nivo: nivolumab; Ipi: ipilimumab; ORR: objective response rate; Pembro: pembrolizumab, PFS: progression-free survival; RFS: recurrence-free survival; Sor: sorafenib; Tis: tislelizumab. ^2^ Status unknown.

**Table 21 cancers-16-01831-t021:** Ongoing studies addressing cytokine-directed therapy in HCC.

Study	Phase	Setting	Target	Treatment	Primary Endpoint	NCT
-	II	palliative	IL-8	BMS-986253	ORR	NCT04050462
-	II	palliative	IL-27	Atezolizumab + Bevacizumab + SRF388	Safety, PFS	NCT05359861
-	I	palliative	IL-27	Pembrolizumab/SRF388	Safety, ORR	NCT04374877
KEYNOTE-D13	II	palliative	IL-15	Pembrolizumab/SOT101	ORR	NCT05256381
-	I	palliative	OX40	HFB301001	Safety	NCT05229601
GDFATHER	II	palliative	GDF-15	Visugromab	Safety, response	NCT04725474

ORR: overall response rate; PFS: progression-free survival.

**Table 22 cancers-16-01831-t022:** Ongoing studies adjuvant treatment of HCC after resection, ablation or LTx.

Study	Phase	Setting	Treatment Arm A	Treatment Arm B	Primary Endpoint	NCT
-	II	adjuvant post LTx	Sor	placebo	1-year RFS	NCT06041490
-	II	LLTHVV	Len	-	3-year RFS	NCT04319484
-	-	LDLT	Len	-	recurrence rate	NCT05572528
CheckMate 9DX	III	adjuvant	Nivo	placebo	RFS	NCT03383458
KEYNOTE-937	III	adjuvant	Pembro	placebo	RFS + OS	NCT03867084
-	I	HCC after liver transplantation	Cam	-	ORR	NCT04564313
JUPITER 04	II/III	adjuvant (resection)	Tor	placebo	RFS	NCT03859128
EMERALD-2	III	adjuvant	Dur + B	Dur + placebo (Arm C: placebo)	RFS	NCT03847428
PREVENT-2	III	adjuvant	Tis + Len	Tis	RFS	NCT05910970
CISLD-8	I	adjuvant	Dona + PD-1	-	RFS	NCT04418401
NEOTOMA	II	adjuvant	Dur plus Tre	-	safety	NCT05440864
DaDaLi	III	adjuvant	Sinti + B	-	RFS	NCT04682210
-	II	adjuvant	Tis + Sitravantinib	-	RFS	NCT05407519
-	II	adjuvant	Dona + Tis	-	RFS	NCT05545124
-	II	adjuvant	Cam + Apa	Cam	DFS	NCT05367687
-	III	adjuvant	Cam + Apa	placebo	DFS	NCT04639180
-	II	adjuvant	Tor	placebo	DFS	NCT05240404
-	I/II	adjuvant	TACE	Len + TACE	Safety, RFS	NCT04911959
-	II	adjuvant	TACE + Don	-	RFS	NCT05161143
ICMJE A	II	adjuvant	TACE + Tis	-	RFS	NCT04981665
ALTER-H006	II	adjuvant	TQB2450(PD L-1) + Anlotinib	-	DFS	NCT05111366
-	II	adjuvant	HAIC + Anlotinib + TQB2450 (4 cycles)	HAIC + Anlotinib + TQB2450 (8 cycles)	DFS	NCT05311319

Apa: apatinib; B: bevacizumab; Cam: camrelizumab; Dona: donatinib; Dur: durvalumab; LDLT: living donor liver transplantation; Len: lenvatinib; LLTHVV: lenvatinib following liver transplantation in patients of hepatocellular carcinoma with portal vein tumor thrombus; LTx: liver transplantation; Nivo: nivolumab; Pembro: prembrolizumab; ORR: overall response rate; OS: overall survival; Sinti: sintilimab; Sor: sorafenib; RFS: recurrence-free survival; Tis: tislelizumab; Tor: toripalimab; Tre: tremelimumab.

**Table 23 cancers-16-01831-t023:** Approved systemic therapies for treatment of HCC.

Study	Phase	Setting	Drug	Target	FDA	EMA	NMPA	NCT
SHARP	III	palliative	Sorafenib	Multi-TKI	X	X	X	NCT00105443
REFLECT	III	palliative	Lenvatinib	Multi-TKI	X	X	X	NCT01761266
CELESTIAL	III	palliative	Cabozantinib	Multi-TKI	X	X	X	NCT01908426
RESORCE	III	palliative	Regorafenib	Multi-TKI	X	X	N/A	NCT01774344
IMbrave150	III	palliative	Atezolizumab/Bevacizumab	PD-L1/VEGF	X	X	X	NCT03434379
HIMALAYA	III	palliative	Durvalumab/Tremelimumab	PD-L1/CTLA-4	X	X	X	NCT03298451
KEYNOTE-224 KEYNOTE-394	II III	palliative	Pembrolizumab	PD-1	X	-	X	NCT02702414 NCT03062358
CheckMate 040	I/II	palliative	Ipilimumab/Nivolumab	CTLA-4/PD-1	X	-	N/A	NCT01658878
STARTRK-1/2	I/II	palliative	Entrectinib	NTRK	X	X	N/A	NCT02097810 NCT02568267
NAVIGATE SCOUT	I II	palliative	Larotrectinib	NTRK	X	X	N/A	NCT02576431 NCT02637687
-	II/III	palliative	Donafenib	Multi-TKI	-	-	X	NCT02645981
RESCUE CARES-310	II III	palliative	Camrelizumab/Apatinib	PD-L1/VEGF	-	-	X	NCT03463876 NCT03764293
ORIENT-32	II/III	palliative	Sintilimab/IBI305	PD-1/VEGF	-	-	X	NCT03794440
RATIONALE-301	III	palliative	Tislelizumab	PD-1	-	-	X	NCT03419897

N/A: not available.

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
