# Peer review of "Insights in Molecular Therapies for Hepatocellular Carcinoma"

_cancers, 2024, doi:10.3390/cancers16101831_

Round 1
Reviewer 1 Report
Comments and Suggestions for Authors
The manuscript submitted by Heumann et al., on Molecular Therapies for Hepatocellular Carcinoma, the manuscript offers a comprehensive analysis of current therapies, providing valuable insights into treating this cancer type. Heumann's thorough review and clear presentation significantly contribute to understanding of hepatocellular carcinoma treatment approaches.
I recommend this manuscript after addressing the minor revisions, will refine the manuscript, ensuring its readiness for publication and further enriching the field.
Kindly find my comments below:
1. The manuscript provides a comprehensive overview of molecular directed therapy for hepatocellular carcinoma (HCC), emphasizing targeted pathways to improve patient outcomes. Recent advancements in tyrosine kinase and angiogenesis inhibition, alongside the significance of VEGF signaling and immune-directed therapy, are highlighted. Kindly highlight few specific studies related to overcome toxicity.
2. The manuscript presents a thorough overview of antibody-drug conjugates (ADCs) as potential therapeutic agents for systemic therapy in hepatocellular carcinoma (HCC). The discussion on the essential components of ADCs and their mechanisms of action is clear and informative. However, to enhance understanding of the practical implications and limitations of ADC therapy in this patient population, including more details on the specific challenges and considerations associated with liver function in clinical trials evaluating ADCs for advanced HCC would be beneficial.
3. The manuscript discusses the potential of CDK4/6 and EGFR inhibitors in treating hepatocellular carcinoma (HCC). While it effectively summarizes preclinical and clinical data, enriching the discussion with a brief mention of resistance mechanisms and combination strategies would enhance its depth.
4. The manuscript also addresses CAR-T cell therapy for liver cancer, mentioning GPC3 and AFP as potential targets and discussing challenges like finding the right targets and dealing with the tumor environment. Adding insights into enhancing CAR-T cell high efficacy and low toxicity challenges would further contribute to the manuscript's value.
5. In the brief overview of vaccination options for liver cancer, promising early results with certain vaccines are noted, alongside discussions on challenges like the body's immune response and the tumor's environment. Suggestions for more research into using viruses for treatment are provided.
6. Furthermore, further exploration of potential therapeutic strategies to overcome resistance mechanisms and enhance the efficacy of MEK inhibitors, including combination therapies or alternative targeting approaches, would be beneficial. Additionally, insights into the clinical implications and future directions of effective tissue targeted delivery to PI3K/AKT/mTOR signaling pathway in HCC treatment could enrich the discussion and provide valuable guidance for future research and clinical practice.
Author Response
Dear Reviewer,
Thank you for your opinion. Please refer to the attached point-to-point revision for our adjustments and suggested changes.
Yours sincerely, Philipp Heumann

Reviewer 2 Report
Comments and Suggestions for Authors
The topic is interesting and the review was nicely written. However, several points should be addressed:
1) The review is way too long!!!! Very hard to read....i suggest to consistently shorten it. For example, sorafenib is the past and it should be commented quickly without spending 3 pages about all the story on the use of sorafenib in HCC patients....
2) I would delete also the first section about the staging....we suppose the reader is aware of the BCLC staging system!
3) The authors should prepare a specific paragraph on the potential adjuvant role of systemic therapies in HCC patients
4) The authors should emphasize a little the comparative results of head-to-head studies among systemic therapies. In this regard, cite and comment the recent SRMA: PMID: 34017396 )
5) The authors commented deeply on the ongoing trials but maybe they should add some comments on the potential prognostic role of several baseline factors (for example etiology, comorbidities....) on patient outcomes
Author Response

(The authors gave the same response as above.)

Reviewer 3 Report
Comments and Suggestions for Authors
I think this is fine but Fig 1 could be a bit larger to be easier to read, and Figure 2 is a bit chaotic and should be simplified. Other than that I saw no other major issues
Author Response

(The authors gave the same response as above.)

Reviewer 4 Report
Comments and Suggestions for Authors
This study summarized the 27 current knowledge for each therapeutic setting and combination that currently is or has been under 28 clinical evaluation. It is interesting. However, the following issues need to be addressed,
1. The preface needs to make a detailed summary of the current situation of molecular therapy for hepatocellular carcinoma, and put forward the purpose and significance of this study; At the same time, in view of the current study the prognosis of hepatocellular carcinoma (HCC) and so on make specific summary, can consider to refer to the following documents: https://doi.org/10.1016/j.compbiomed.2024.108260;
2. The study needs a general flow chart
3. How many stages has the development of molecular therapy gone through? It is recommended to make a time development chart
4. Some important molecular therapy models need to be elaborated and put forward their own insights as much as possible
5. What did the review of this research give you? What is your vision for the future? It needs to be clarified in the conclusion.
Comments on the Quality of English Languagenone
Author Response

(The authors gave the same response as above.)

Round 2
Reviewer 2 Report
Comments and Suggestions for Authors
The revised version of the manuscript is OK. Thank you!